



# Statistical characterization of roughness uncertainty and impact on wind resource estimation

Mark Kelly[1] and Hans Ejsing Jørgensen[1]

[1]Wind Energy Division/Meteorology Section, Risø Lab./Campus, Danish Technical University; Roskilde 4000  Denmark.
*Correspondence to:* Mark Kelly (`mkel@dtu.dk`)

**Abstract.**

In this work we relate uncertainty in background roughness length ($z_0$) to uncertainty in wind speeds, where the latter are predicted at a wind farm location based on wind statistics observed at a different site. Sensitivity of predicted winds to roughness is derived analytically for the industry-standard European Wind Atlas method, which is based on the geostrophic drag law. We consider roughness statistically and its corresponding uncertainty, in terms of both $z_0$ derived from measured wind speeds, as well as that chosen in practice by wind engineers. We show the combined effect of roughness uncertainty arising from differing wind-observation and turbine-prediction sites; this is done for the case of roughness bias, as well as for the general case. For estimation of uncertainty in annual energy production (AEP), we also develop a generalized analytical turbine power curve, from which we derive a relation between mean wind speed and AEP. Following from our developments we provide guidance on approximate roughness uncertainty magnitudes to be expected in industry practice, and also find that sites with larger background roughness incur relatively larger uncertainties.

## 1   Introduction

Microscale flow models have been employed for decades in wind energy assessment, for estimation of resources at one location based on wind measurements at a different site (Troen and Petersen, 1989). Further, it has become increasingly popular in the past decade to use mesoscale model output to drive microscale models, for the same purpose (e.g. Badger et al., 2014). Such flow modeling relies on characterization of the surface, including terrain elevation and surface roughness. As input to atmospheric flow models, both terrain elevation and roughness have uncertainties associated with their assignment. In practice, terrain elevation uncertainty tends to be dominated by the resolution of elevation maps (e.g. Sørensen et al., 2012).[1] In contrast, there are a number of significant uncertainties associated with roughness, which do not (necessarily) depend on resolution; these include determination of roughness length $z_0$ from measurements, and assignment of $z_0$ in industrial practice (based on e.g. land-use/terrain type and/or experience). Overall, uncertainty related to roughness tends to be dominant over elevation-related

---

[1] Currently (2016), microscale models typically have computational resolutions finer than elevation maps; commonly available elevation maps in most of the world today have typical resolutions of $\Delta x \sim 10$–$90$ m, whereas quasi-linear (e.g. WAsP) and Reynolds-averaged Navier-Stokes (RANS) models employed for wind are most often run with resolutions (much) finer than 10 m. There are a growing number of exceptions, stemming from the advent of airborne laser-based terrain measurements which can offer resolutions less than one meter (e.g. Zhang et al., 2005; Danish Geodata Agency, 2015).





uncertainty, particularly in wind-energy applications. In this work we develop a practical treatment of the effect of roughness uncertainty upon wind resource estimation, providing a formulation for estimation of roughness-induced uncertainty in annual energy production.

First we review the definition of roughness length, introducing and demonstrating the statistical character of $z_0$—i.e. distributions of $z_0$ from measurements, and the behavior of such; we connect this statistically to a practical uncertainty metric. Then we present the theoretical framework which is used for wind resource estimation, based on the geostrophic drag law (as used in the European Wind Atlas methodology, Troen and Petersen, 1989) and including its relation to roughness. In section 3 we introduce uncertainty; this includes basic characterization of the uncertainties inherent in [1] the roughness definition and observed distributions of $z_0$ (§3.1.1), and [2] the variations in $z_0$ prescribed in the wind energy industry (3.1.2). We continue by showing how uncertainty in the background roughness can be translated into uncertainty in predicted wind distributions, within the European Atlas framework (§3.2.1); here we provide derivations of the sensitivity of predicted winds to input roughnesses at observation and prediction sites, respectively. Consequently we examine the effect of user-assigned biases in roughness assignment, and more generally the combined effect of (independent) roughness uncertainties on predicted wind speeds. For practical use we also develop an analytical relation between rated power, mean wind speed (Weibull-$A$ parameter), and AEP; this is accomplished via convolution of a generalized analytical power curve form and Weibull wind distribution. Thus we translate $z_0$-uncertainty into uncertainty of annual energy production [AEP].

Though there are different methods possible for determining or calculating roughness length, we concentrate here on the propagation of uncertainty in background roughness to predicted wind speeds and annual energy production. More details about and issues arising from alternate methods of roughness length calculation are beyond the scope of this article, and are the basis of concurrent work to be included in a separate paper(s).

Lastly we discuss approximate roughness uncertainty magnitudes expected in practice, and the consequences of such. This also includes, for example, the result that sites with larger background roughness tend to give larger relative uncertainty (i.e. %) in predicted wind speeds and significant uncertainty in AEP. We also discuss implications on the use of mesoscale simulation data for driving microscale models, i.e. generalization of wind statistics.

## 2   Basis and framework

Physically, this work simply considers the use of wind measurements (statistics) at some height above ground level at one location, in order to predict wind statistics at another location and height. Starting with ideal (uniform flat) terrain, this prediction can be broken into components, commonly labeled within the wind resource assessment community as vertical and horizontal 'extrapolation,' respectively. Subsequently the theoretical foundation of this work involves the two basic components related to the physics modeled by such 'extrapolations': these are the wind profile for vertical extrapolation, and the geostrophic drag law ('GDL') for relating the wind statistics at different sites; they are covered in sub-sections 2.1 and 2.2, respectively. The vertical wind profile form (of which the simplest is the logarithmic law) requires a surface roughness length, and the GDL also requires a characteristic (background) roughness length. Because we wish to relate uncertainty in roughness to uncertainty in





wind energy estimates, i.e. finding the uncertainty in accounting for the effect of the surface—we first begin by examining roughness length, both in theory (i.e. definition) and in practice (e.g. its statistical character).

## 2.1 Roughness length: theory and practice

The concept of roughness length began with characterization of the velocity profile in ideal engineering flows (e.g. pipes),
where roughness has a direct physical interpretation (Nikuradse, 1933; Tripp, 1936); it was further adopted to describe the wind profile in the atmospheric surface layer (ASL), whereby it has an implicit (and not directly physical) definition (Monin and Yaglom, 1971). The basic role of roughness length, and its definition, can be seen through the ideal expression for the mean wind profile $U(z)$ over a homogeneous flat surface in neutral conditions (without thermal stability effects):

$$U(z) = \frac{u_*}{\kappa} \ln\left(\frac{z}{z_0}\right). \tag{1}$$

In (1) $z_0$ is the roughness length and $z$ the height above (distance normal to) the surface, expressed in the same units; $\kappa$ is the von Kármán constant, generally accepted to be 0.4 (Högstrom, 1996). The friction velocity $u_*$ is defined by $u_*^2 \equiv -\langle u'w' \rangle$, i.e. as mean momentum transport towards the surface through turbulent stream-wise ($u'$) and vertical or surface-normal ($w'$) velocity fluctuations. The roughness $z_0$ can also be seen as an integration constant, since (1) results from integrating $dU/dz = u_*/(\kappa z)$; the latter is typically derived via dimensional analysis, through the Buckingham Pi theorem (e.g. Stull, 1988; Wyn-
gaard, 2010). The logarithmic wind profile (1) depends upon a number of assumptions: $u_*$ is effectively constant from the surface up to height $z$ and (i.e. $du_*^2/dz \ll$), the surface is flat and uniform, there is horizontal homogeneity (no variations parallel to the surface), there is no height-dependence in the forcing of the flow, and there are (no effects due to) temperature variations; i.e. the only variables determining $dU/dz$ are $u_*$ and $z$.

### 2.1.1 Calculation of roughness length from wind measurements

From (1) one can see that for $U$ measured at two heights $\{z_1, z_2\}$, the roughness can be calculated by

$$\ln(z_0) = \frac{U(z_2)\ln z_1 - U(z_1)\ln z_2}{U(z_2) - U(z_1)}. \tag{2}$$

While one can also obtain the roughness via the shear exponent (e.g. Kelly et al., 2014a) that is oft-used in wind energy, Equation 2 does not involve approximations, and directly follows from the definition of roughness. One can also use friction velocity measured in the surface layer and wind speed from one (or more) height(s) to derive roughness (e.g. $z_0 = z\exp\left[-\kappa U(z)/u_*\right]$
from Eqn. 1), but doing so requires sonic anemometers, which are not yet commonly used in the wind energy industry. Thus we use (2) for the 'observed' roughness data analyzed and shown in this paper, and leave alternate $z_0$-estimation methods for concurrent work/dissemination that focuses solely upon roughness. This choice is further supported by the focus of the present article—we are concerned here with the impact of roughness length on wind energy estimates—and because we develop and use an uncertainty-estimation framework that is generally applicable to $z_0$, regardless of whether $z_0$ is derived from (2) or
via $U/u_*$.




### 2.1.2 Roughness as a *statistic*

Even in seemingly ideal conditions—such as measuring wind profiles in the surface layer at a site where the terrain is flat and appears uniform, with non-neutral cases excluded—in practice one still observes a broad range of roughnesses. This is demonstrated in Figure 1, which shows the roughness length calculated from 10 m and 40 m measurements at the Danish

5  National Wind Turbine Test Station at Høvsøre, for upwind directions corresponding to flat and homogeneous surface (east of the meteorological measurement mast). Here we have filtered out non-neutral conditions by keeping only cases unaffected by thermodynamic stability by using $z/|L| < 0.001$, i.e. for Obukhov lengths $L$ much greater than the heights of measurement. Figure 1 starkly demonstrates that, even at a "homogeneous," well-studied and presumably simple site, roughness length has

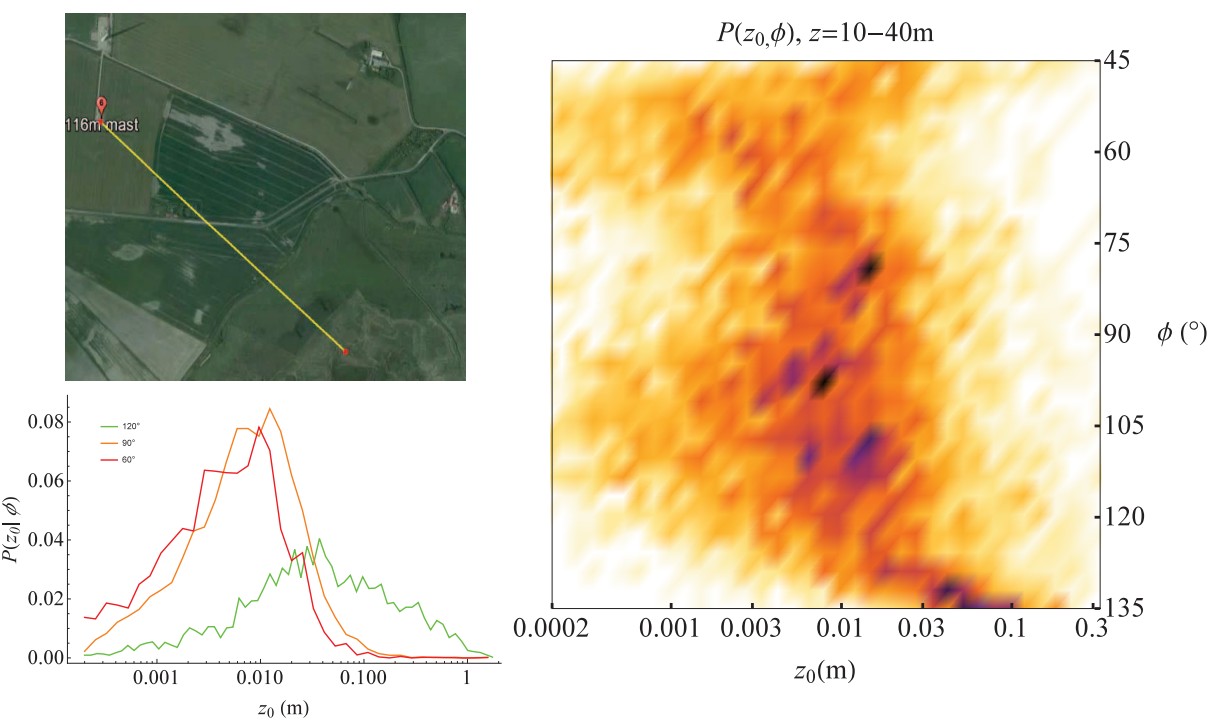

**Figure 1.** Lower-left: Distribution of $z_0$ for 'homogeneous' land sectors (30° wide) east of Høvsøre. Right: Joint distribution of $z_0$ and (wind) direction; darker represents most common values, white is no occurrence. Calculation follows (2) with $z_1 = 10$ m and $z_2 = 40$ m, and is limited to neutral conditions ($|L|^{-1} < 0.001$ m$^{-1}$). Upper-left: visual map east of site (red pointer; southern border of 'homogeneous' zone at ~130° denoted by yellow line).

a distribution of significant width. Note that we plot the distribution of roughness length in logarithmic space; this is done

10  because it is $\ln(z_0)$ which directly affects the wind profile, as in (1). This also highlights the breadth of the distribution (several orders of magnitude), and that we must subsequently approach roughness uncertainty in a *multiplicative* (dimensionless) way and not in an additive way. We also remind that the roughness lengths generally used in wind flow modeling and resource assessment actually correspond to some *geometric* mean, which *should* be based on the $z_0$-distribution (alternately one can





express wind profiles in terms of the distribution $P(\ln z_0)$ and corresponding arithmetic mean, c.f. Kelly and Gryning, 2010); unfortunately $z_0$ are not (yet) defined explicitly as such in typical wind engineering practice. We thus focus in this paper on roughness uncertainty within the 'implied mean-roughness' framework implicit in standard wind engineering.

In addition to the relatively wide distribution apparent for roughnesses obtained from 30-minute averages shown in Fig. 1

(and slightly wider for 10-minute averages, not shown), one can also see some local—and non-ideal—details. One sees the minor effects of a barn and a small building located roughly 800 m upwind at $\sim 80°$ and $\sim 110°$ respectively, as well as the larger effect of the seasonally-varying marsh/fjord coastline 800–900 m to the southeast ($\sim 130$–$135°$). Such roughness changes tend to violate the assumptions behind the logarithmic profile, over a range of observation heights falling within the non-equilibrium internal boundary-layer (IBL) transition region (Semprevira et al., 1990; Bou-Zeid et al., 2004; Calaf et al.,

2014).[2] The more drastic semi-coastal roughness change 'contaminates' the shear measured between 10–40 m enough to give the larger apparent $z_0$ shown in Fig. 1a as $\phi \rightarrow 135°$ and subsequently wider distribution $P(z_0)$ shown in Fig. 1b for the $120°$ sector.

Because neutral conditions tend to be encountered most often (stability distributions have their peak around $L^{-1} = 0$, c.f. Kelly and Gryning, 2010), the distribution of shear exponent $P(\alpha)$ can also be related in terms of an effective rough-

ness length without filtering stability to exclude non-neutral conditions (Kelly et al., 2014a). Thus the wind profile can indeed give information about the surface, though the shear at higher $z$ includes the effect of increasingly more terrain further upwind (potentially including hills as well as roughnesses).[3] Avoiding substantial changes in surface characteristics/land use, this can be useful towards the aim of gauging background $z_0$.

One can also calculate a more 'local' roughness length via (1) using measurements of $U$ and $u_*$ within the surface layer

(filtering out non-neutral conditions via measured heat fluxes), but doing so requires sonic anemometers, which are not (yet) commonly used in the wind industry. For example, using $U$ and $u_*$ measured at $z$=10m for the case above gives $P(z_0, \phi)$ that is insensitive to the inhomogeneities described above, i.e. it does not 'jump' as $\phi$ increases above $\sim 130°$. But although the resultant $z_0(U/u_*)$ tend to better conform to the assumptions behind surface-layer theory and (1), they are consequently limited to ASL heights—which in stable conditions (e.g. nighttime, winter) only extend to $\sim 10$–$20$ m. Further, the $z_0$ derived from

$U/u_*$ in the ASL are local, only pertaining to the nearest several hundred meters, perhaps less in stable conditions. However, the widths of $P(\ln z_0)$ derived from $U/u_*$ (not shown) are on par with those obtained from $U$ at two heights and displayed in Fig. 1.

Thus in the present article concerned about uncertainty, we do not address the implications of surface-layer theory nor its conditional violation, but rather focus on the effect of roughness uncertainty—as it would be measured (or assigned) in

industrial practice—upon resource assessment, particularly through 'horizontal extrapolation' from an observation mast to a separate turbine location(s).

---

[2]The IBL develops downwind from a roughness change with expansion slope ($z$:$x$) of roughly 1:100, and the top of the associated transition region expands at a variable rate of 1 to $\sim$12–15. For the example noted here this corresponds to the flow measured by anemometers at both 10 m and 40 m being affected.

[3] The increasing area of surface affecting winds at increasing heights, and also associated averaging issues, are beyond the scope of the current article (consult e.g. Lettau, 1969; Garratt, 1978; Hasager and Jensen, 1999).





## 2.2 Geostrophic drag law: Wind Atlas method

The geostrophic drag law (GDL) allows wind statistics observed at one site to be applied at potential wind farm sites nearby that may have different surface characteristics (i.e. roughness and terrain elevation); it is the basis of the European Wind Atlas ['EWA'] Method (Troen and Petersen, 1989) used widely for wind resource estimation. The GDL arises from matching the dimensionless surface-layer profile of mean wind in neutral conditions (i.e. the log-law divided by $u_*$) to dimensionless solutions of the mean horizontal equations of motion away from the surface, as affected by the Coriolis force (Ellison, 1956; Krishna, 1980; Walmsley, 1992). The mean atmospheric boundary layer (ABL) flow is driven by a large-scale mean pressure gradient $\nabla P$, expressible also as the geostrophic wind $\mathbf{G} \equiv -\hat{\mathbf{k}} \times \nabla P / (f\rho) = \{-\partial P/\partial y, \partial P/\partial x\}/(f\rho)$ where $\hat{\mathbf{k}}$ is the vertical unit vector and $f$ is the latitude-dependent Coriolis parameter; the pressure gradient force is balanced (vectorially) by the Coriolis force and momentum transfer to the surface. So the GDL basically relates the large-scale forcing (expressible as the geostrophic wind above the ABL) to the surface-layer momentum flux (friction velocity), depending on the surface roughness.

The geostrophic drag law can be simply expressed in scalar form as

$$G = \frac{u_{*0}}{\kappa} \sqrt{\left[ \ln\left( \frac{u_*/f}{z_0} \right) - A_0 \right]^2 + B_0^2},$$
(3)

where $A_0$ and $B_0$ are empirical constants (taken e.g. by the EWA to be 1.8 and 4.5, respectively). Thus for two sites which can be assumed to have the same large-scale forcing (distribution of $\mathbf{G}$), then the wind statistics at one site can be translated to wind statistics at the other. From the wind profile relation (1) one can obtain $u_*$ from measured $U$ over one roughness $z_{0,1}$, and subsequently $G$ from (3); then at the prediction site one can solve (3) to get $u_*$ at a potential turbine site, and subsequently find $U$ there over a roughness $z_{0,2}$. Below, we will show the impact of roughness uncertainty upon wind speed and AEP estimates via (1) and (3).

## 3 Uncertainty

### 3.1 Roughness and uncertainty components

In general, uncertainty can be classified into two types (Kiureghian and Ditlevsen, 2009): *aleatoric* uncertainty, and *epistemic* uncertainty. First, *aleatoric* (sometimes called "statistical" or "random") uncertainty is the variability in a quantity that arises from randomness inherent the process(es) which impact said quantity. *Epistemic* or "systematic" uncertainty is that which arises due to lack of knowledge about a quantity (imperfect understanding of it in the 'real world').

The aleatoric (random) uncertainty inherent in roughness length can be said to include that associated with the width of the 'observed' distribution of $z_0$ shown in section 2.1.2. This tends to be due to variability in the system being described; the 'system' in this case is the atmospheric surface layer and the surface nearby the measurement point which influences the flow. However, there is also an epistemic component containted within the distributions $P(z_0)$ shown in Figure 1; it is due to effects which were neglected in the derivation of the theory used, namely the logarithmic law (1). Physically, this includes inhomogeneities in the surface upwind, and dependence of surface characteristics upon wind speed (i.e. water or flexible



vegetation, c.f. Monin and Yaglom, 1971); within the context of the turbulent surface layer as described by turbulence theory, it tends to be manifested via turbulent transport (Kelly et al., 2014a; Sogachev and Kelly, 2015).

When performing resource assessment, wind engineers in practice characterize the surface via roughness length (as well as terrain elevation, which we do not treat in this paper). Roughness characterization can occur via assignment of $z_0$ values
chosen by the wind engineer, or through roughness values (or land-use types) 'inherited' from maps acquired from a third party. Typically the former has dominated the wind industry, though the latter is becoming more common; land-use types and classes are contained in some geographical data products, but these have not yet been shown to be consistently or universally translatable to roughness lengths for different parts of the world (see e.g. Marticorena et al., 2006; Torbick et al., 2006). Either way, *epistemic* uncertainty arises due to our ignorance of the appropriate representative roughness length[4] and is introduced
when characterizing the surface via a single roughness; this uncertainty exists regardless of whether the characteristic $z_0$ is chosen by an algorithm assigning values to a map based on look-up tables for various land classifications, or by a wind engineer who has visited the (potential) site.

The epistemic components associated with the theory used to 'convert' wind observations into 'observed' $z_0$ tend to manifest via turbulent transport, and subsequently behave randomly, arising to a good degree from variability of the surface itself (hence
being debatably aleatoric). These are in contrast to the uncertainty arising from selection of $z_0$ by engineers, or the uncertainty inherent in (usage of) a relatively small number of widely-used sources for roughness-maps—which can contain significant bias and are not (directly) related to measurement. Thus here we group the former, observationally-related uncertainty together with the aleatoric uncertainty, and then consider separately the epistemic uncertainty implicit in assignment of roughness values by wind engineers in practice.

### 3.1.1  Uncertainty in observation-based $z_0$

For the observation-based roughness lengths displayed in Section 2.1.2 (Fig. 1), the distributions are best described (and thus plotted) as $P(\ln z_0)$—again consistent with both the $\ln(z_0)$ behavior expected within the wind profile, and with the geometric (multiplicative) averaging needed to obtain a characteristic 'mean' roughness. The width of the $\ln(z_0)$ distributions shown in Fig. 1 gives indication of the *variability* in $\ln z_0$ over many 30-minute (or 10-minute) periods. In particular the $P(\ln z_0)$ for
$30°$-wide directional sectors can be considered, that is $P(\ln z_0|\varphi)$, since sectors of this width are commonly used in resource assessment. The homogeneous $60°$ and $90°$ sectors at Høvsøre (Fig. 1) have similar shapes, and both exhibit half-peak widths of roughly one-half order of magnitude (a factor of $\sim 3$); i.e. for a given sector's background roughness $z_0$, the width of the distribution can be seen as that defined roughly between $z_0/3$ and $3z_0$.

However, the uncertainty in determining a representative roughness length—via the appropriate (geometric) mean—is not
the same as the width of the $\ln(z_0)$ distribution. Rather, the uncertainty in the mean roughness is the width of the distribution of expected means calculated for a given site and sector. For this purpose we use a basic 'bootstrap' resampling method (Varian, 2005; Wu, 1986): simply re-sampling randomly from the diagnosed (30-minute) roughness lengths, we synthesize a distribution

---

[4] As shown in the section above, the representative roughness length should be based on a *geometric mean*, due the $\ln(z_0)$ behavior exhibited by the surface-layer wind profile in neutral conditions.





of $10^5$ values of geometric-mean $z_0$ per sector. This results in a log-normal distribution of mean $z_0$ (Gaussian distribution of $\ln z_0$); this distribution $P(\exp[\langle \ln z_0 \rangle])$ is centered around a value equal to the geometric mean that had been found for each sector by operating directly on the wind data. The width of each (sector-wise) distribution of mean-roughnesses from resampling depends on the number of re-sampled points used to create each mean in the synthesized distribution. For a number

equivalent to one year's worth of data (based on the sector-wise frequency of occurrence), the mean-distributions are in fact much narrower than the distributions shown in Figure 1. The bootstrapped mean-roughness distribution is almost perfectly fit by a log-normal form; the half-width $w_{\langle z_0 \rangle_{\mathrm{RS}}}$ for this form can be simply expressed non-dimensionally (i.e. effectively normalized by the expected mean) via the standard deviation of mean-$\ln z_0$ from resampling ($\sigma_{\langle \ln z_0 \rangle_{\mathrm{RS}}}$), as

$$\frac{w_{\langle z_0 \rangle_{\mathrm{RS}}}}{\left\langle \langle z_0 \rangle_{\mathrm{RS}} \right\rangle_g} = \exp \left\{ \sigma_{\langle \ln[z_0 / \langle z_0 \rangle_g] \rangle_{\mathrm{RS}}} \right\} = \exp \left\{ \sigma_{\langle \ln z_0 \rangle_{\mathrm{RS}}} \right\} - 1. \tag{4}$$

For the Høvsøre 'homogeneous' land sectors treated here and the bootstrapped means each calculated from one-year's worth of resampled data, the $w_{\langle z_0 \rangle_{\mathrm{RS}}}$ of the sector-wise distributions of these means are about 5% of the expected mean roughness length (specifically, 5.4%, 4.1%, and 5.3% of the respective $\langle \langle z_0 \rangle_{\mathrm{RS}} \rangle$ in each sector from Eq. 4). Thus, considering only calculation of $z_0$ from wind speeds measured at two (10 and 40 m) heights via (2) from one year of data, the roughness uncertainty for the three sectors shown in Figure 1 is about 5%. For longer data sets, the uncertainty decreases; for example, randomly drawing

from the entire 10-year set leads to half-widths of 1–2%.

We do remind that there are other methods to calculate $z_0$, such as using the surface-layer friction velocity $u_*$ and wind speed at one (or more) measurement height(s) via (1)—which may result in different values of estimated mean/characteristic roughness length. For example, repeating the analysis above using (1) with $U$ and $u_*$ measured at 10 m height, we again obtain well-behaved distributions of bootstrapped mean roughness whose half-widths are about 5%; one might take this as the implied

uncertainty. However, the mean values (for a given sector) can actually differ between the two methods, by an amount which can greatly exceed 5% (in these Høvsøre land sectors they can differ by a factor of $\sim$3!). This difference is related to the flow physics at increasing distances from the mast (the momentum flux footprint), the details of which are beyond the scope of this paper; we defer further discussion of such differences to section 4.

### 3.1.2 Uncertainty and ensembles of user-input

Even for an ideal homogeneous landscape, the wind industry, which is a collection of wind engineers and companies, will *as a group* assign different roughnesses to characterize the surface (whether actively or inherited via acquired maps). This results in a distribution of $z_0$ assigned to predict the wind for any given site, and in effect to an (epistemic) uncertainty—and subsequently industry-wide variation in predicted AEP, even at the most simple sites.

We provide a simple practical example of gauging such epistemic uncertainty, based on a systematic exercise: we asked

separate groups of wind resource assessment experts to individually evaluate the surface roughness length for two commonly-encountered land surface types. The groups of participants in this exercise were polled at meetings of the Danish Windpower Network 'Vindkraft-Net' (Kelly and Jørgensen, 2014) and of the Meteorology section of the Department of Wind Energy (Risø lab/campus) in the Danish Technical University, respectively; their work and foci range from wind engineering and



commercial site assessment to research in boundary-layer meteorology and wind resource calculation. The participants were

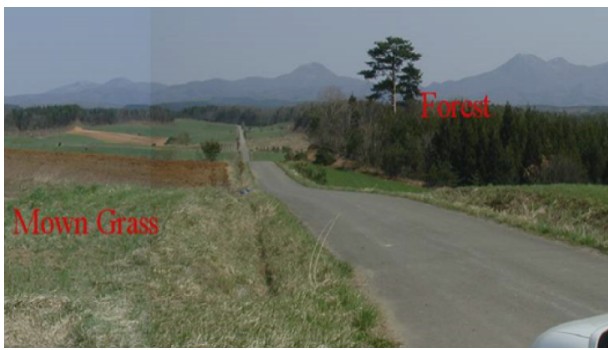

**Figure 2.** Image of the two areas (grassy and forested) used in roughness survey exercise.

shown a picture containing both a grassy area and a forested area (the latter specified as having a mean tree height of 15 m), and asked to give $z_0$ for each of these two areas; the picture is replicated in Figure 2. The "raw" results of the roughness survey, which consisted of 19 and 28 participants, respectively, are shown in Table 1.

| Group | geom. mean, $\langle z_0 \rangle_g$ | | arith. mean, $\langle z_0 \rangle$ | | $\exp\{\sigma_{\ln[z_0/\langle z_0 \rangle_g]}\}$ | | std. deviation, $\sigma_{z_0}$ | | $n$ |
|---|---|---|---|---|---|---|---|---|---|
| | Grass | Forest | Grass | Forest | Grass | Forest | Grass | Forest | |
| Vindkraft-net | 4.0 cm | 0.87 m | 5.6 cm | 1.6 m | 124% | 162% | 6.4 cm (115%) | 2.5 m (158%) | 28 |
| DTU Wind | 4.2 cm | 0.82 m | 5.5 cm | 1.0 m | 112% | 113% | 4.7 cm (86%) | 0.57 m (57%) | 19 |
| Combined | 4.1 cm | 0.85 m | 5.6 cm | 1.3 m | 117% | 141% | 5.7 cm (103%) | 2.0 m (146%) | 47 |

**Table 1.** Means (geometric and arithmetic) and corresponding deviations in $z_0$, surveyed from two groups of wind resource experts for the two terrain types shown in Fig. 2. For conventional (linear) standard deviation $\sigma_{z_0}$, number in parenthesis is $\sigma_{z_0}/\langle z_0 \rangle$, given for comparison with the logarithmic standard deviation ($\exp\{\sigma_{\ln[z_0/\langle z_0 \rangle_g]}\}$).

Note that Table 1 includes not only a geometrically-defined mean $\langle z_0 \rangle_g \equiv \left[\prod_i^n (z_0)_i\right]^{1/n} = \exp\left[\sum_i^n \ln(z_0)_i/n\right]$ and associated dimensionless standard deviation $\exp\{\sigma_{\ln[z_0/\langle z_0 \rangle_g]}\}$ that are consistent with the logarithmic definition of roughness, but also the commonly-used arithmetic mean and (normalized) standard deviation of user-estimated $z_0$. The latter statistics are included for comparison, and because (in contrast to the flow physics) there is some tendency for wind engineers to 'think linearly' rather than logarithmically. As can be seen in Table 1, the arithmetic (linear) mean of $z_0$ is unsurprisingly larger than the properly (logarithmically) averaged $z_0$, by ∼30–40% for grass and ∼20–80% for forest. Arithmetic calculation of $z_0$ statistics subsequently tends to give a smaller normalized deviation compared to the proper log-rms statistic for the raw surveyed data, particularly as the $z_0$-distribution is dominated by values smaller than 1 m (expected from the mathematical character of geometric [$\ln z_0$] versus arithmetic averages). Overall the variability in polled roughness lengths for the two cases is on the





order of but larger than the expected roughness length itself, i.e. by a factor of $\sim$1.1–1.3 times the estimated mean $z_0$ for grass or $\sim$1.1–1.6 times the mean for the forest case. This might be taken as an estimate for uncertainty in $z_0$ for such cases.

The variability in the user data differs between the polled groups, and might be affected by the limited sample size. Due to the limited distributions of polled roughness lengths (not shown) gathered from each of the two expert groups, an alternate

estimate of collective user-uncertainty (i.e. industry-wide) is provided by again applying a resampling method to the distribution of surveyed $z_0$. Following the averaging of expert-elicited $z_0$ and the uncertainty characterization of the previous section, 'bootstrapping' (Varian, 2005; Wu, 1986) is used to resample the elicited $z_0$ values and construct a distribution of the means. Calculating each mean from $n$ non-unique random data samples and repeating $10^6$ times, we generate distributions of $\overline{z_0}$ for the two cases. For $n \gtrsim 3$ we find log-normal distributions for the bootstrapped geometric mean ($\overline{z_0} = \langle z_0 \rangle_g$), as expected

from the central limit theorem ($\ln \overline{z_0}$ becomes Gaussian). In the limit of the sample data set being perfectly representative of wind industry practices, the bootstrapped distribution for a given $n$ is equivalent to the $P(\overline{z_0})$ expected when any given wind engineer uses $n$ values to calculate the mean roughness for a site such as the grass or forest case used here. The means of the resampled distributions are the same as for the raw roughness samples in Table 1, regardless of $n$. The deviation, however, decreases with $n$. For $n = 1$ the deviations converge to those in in Table 1, while the values of the effective geometric

deviation $w_{\langle z_0 \rangle_{RS}}$ behave as approximately $1 + n^{-0.53}$ (the deviations fall slightly more rapidly than $n^{-1/2}$ due to the slightly irregular sample/survey). As an example, Table 2 shows the geometric means and deviations for these mean-distributions using $n = 3$, for the two groups and cases considered.

From Table 2 one infers the seemingly obvious result that, for users taking an average of three 'industry-accepted' roughness

| | geom. mean, $e^{\langle \ln \langle z_0 \rangle_{g,RS} \rangle_g}$ | | eff.dev., $w_{\langle z_0 \rangle_{RS}}$ | |
| Group | Grass | Forest | Grass | Forest |
|---|---|---|---|---|
| Vindkraft-net | 4.0 cm | 0.87 m | 58% | 73% |
| DTU Wind | 4.2 cm | 0.82 m | 53% | 53% |
| combined | 4.1 cm | 0.85 m | 56% | 65% |

**Table 2.** Bootstrapped statistics of mean roughnesses, from (resampled) user-provided $z_0$ given by two groups of wind resource experts, using 3 resampled values per mean calculation; data is for the two terrain types shown in Fig. 2.

estimates (assuming those span the sample taken)—instead of just one—the expected (industry-wide) uncertainty is reduced;

we point out that such a conclusion depends on having reasonably representative roughness values to choose from.

To summarize, in this subsection we saw that the equivalent (normalized logarithmic) standard deviation from surveys of engineer/user-assigned roughness is of the order of the expected roughness itself, as shown in Table 1. In terms of (9), we expect an uncertainty equal to the half-width of the (expected user input) distribution of $\ln z_0$, to be approximately $w_{\langle z_0 \rangle_g} \sim \langle z_0 \rangle_g$. In the following section we would like to show, in general, how uncertainty in $z_0$—whether due to user-input or measurement—

propagates into wind speed and AEP estimates.





## 3.2 Propagation of roughness uncertainty

The uncertainty in roughness length has an effect on a number of key variables needed for wind resource assessment. Since the geostrophic wind depends upon the surface friction velocity $u_*$, in practice one must use a wind profile form (model) to translate measured wind statistics (e.g. Weibull-$A$ or mean wind speed) into the corresponding $u_*$-analogue. This is typically

accomplished by using the log-law (1), which is valid in statistically neutral conditions, and approximately 'in the mean' (Kelly and Gryning, 2010; Kelly and Troen, 2016). Further, to relate $u_*$ at the prediction site to the (mean) geostrophic wind $G$, equation 3 must somehow be solved for $u_*$. A direct analytical solution for $u_*(G)$ via (3) is not possible, so Jensen et al. (1984) developed the approximate "reverse geostrophic drag-law" form

$$u_{*G} = \frac{0.485G}{\ln(G/fz_0) - A_0}.$$ (5)

We adopt (5) and use it along with (1) and (3) in order to relate wind speeds and roughness lengths for a given pair of prediction- and measurement-sites.

### 3.2.1 Sensitivity of predicted wind speed to background roughnesses

By using the logarithmic wind profile (1) at both measurement and prediction locations, along with the 'forward' and 're-verse' geostrophic drag-law forms (3) and (5), one can write the predicted wind speed $U_{\text{pred}}$ in terms of the prediction-site

roughness $z_{0,2}$ and geostrophic wind $G$. The geostrophic wind is further expressible in terms of the measured wind $U_{\text{obs}}$, measurement height $z_{\text{obs}}$, and background roughness $z_{0,1}$ for the measurement site. The resulting expression for $U_{\text{pred}}$ can be differentiated with respect to any of $\{U_{\text{obs}}, z_{\text{obs}}, z_{\text{pred}}, z_{0,1}, z_{0,2}\}$, in order to find the sensitivity of predicted wind speed $U_{\text{pred}}$ to these quantities. We would like to know the effect of roughness uncertainty upon $U_{\text{pred}}$; taking its derivative with regard to the roughness lengths at observation and prediction heights, and re-arranging, we obtain the useful expressions

$$\frac{\partial \ln U_{\text{pred}}}{\partial \ln z_{0,1}} \simeq$$

$$\frac{1}{\ln(z_{\text{obs}}/z_{0,1})} \left[1 - \frac{1}{\ln[G/(fz_{0,2})] - A_0}\right] \left\{1 - \left[\frac{U_{\text{obs}}/G}{\ln(z_{\text{obs}}/z_{0,1})}\right]^2 \left[\ln\left(\frac{\kappa U_{\text{obs}}}{fz_{0,1}}\right) - A_0\right] \left[\ln\left(\frac{z_{\text{obs}}}{z_{0,1}}\right) - 1\right]\right\}$$ (6)

and

$$\frac{\partial U_{\text{pred}}}{\partial \ln z_{0,2}} \simeq \left(\frac{c_G G}{\kappa}\right) \frac{A_0 + \ln\left(z_{\text{pred}} f/G\right)}{\left[\ln\left(G/(fz_{0,2})\right) - A_0\right]^2}.$$ (7)

Here we have made the expression compact, by writing $G(U_{\text{obs}}, z_{\text{obs}}, z_{0,1})$ simply as $G$. Inspection of the two sensitivity expressions (6) and (7) reveals that $U_{\text{pred}}$ is more sensitive to the background roughness at the observation site ($z_{0,1}$) than the

roughness $z_{0,2}$ at the prediction site. Further, it is seen that $U_{\text{pred}}$ also has some sensitivity to observation height $z_{\text{obs}}$, while $z_{0,1}$ dominates.





From (A1)–(A2), which follow from equations 6–7 (see Appendix A for details), we arrive at an (implicit) expression relating the uncertainty in predicted hub-height wind speed to the uncertainty in background roughness at the observation site ($\Delta z_{0,1}$):

$$\left.\frac{\Delta U_{\mathrm{pred}}}{U_{\mathrm{pred}}}\right|_{\Delta z_{0,1}} \simeq \exp\left\{1.1\left(1+\frac{z_{\mathrm{obs}}}{80\,\mathrm{m}}\right)^{-1/7}\left[\mathrm{li}\left\{(z_{\mathrm{obs}}/a_1 z_{0,1})^{-1/7}\right\}-\mathrm{li}\left\{(z_1/z_{0,1})^{-1/7}\right\}\right]\right\} \tag{8}$$

where $\mathrm{li}(x)$ is the log-integral function (e.g. Abramowitz and Stegun, 1972, see appendix also). In (8), $a_1$ is the fractional
uncertainty in observation-site background roughness length,

$$a \equiv \frac{z_0+\Delta z_0}{z_0} = 1+\Delta z_0/z_0 \tag{9}$$

evaluated at $z_0 = z_{0,1}$. Thus roughness uncertainties can be described geometrically (as they should be): for a given background roughness we then have a range of log-roughness described by $\ln(z_{0,1})\pm\ln(a)$, corresponding roughness lengths ranging from $z_{0,1}/a$ to $az_{0,1}$.

Just as (8) was derived above for variations in roughness at the measurement site, we similarly derive the uncertainty in predicted wind speed due to uncertainty in the prediction-site roughness $z_{0,2}$, from (7):

$$\left.\frac{\Delta U_{\mathrm{pred}}}{U_{\mathrm{pred}}}\right|_{\Delta z_{0,2}} \simeq \left[1-\frac{\ln a}{\ln(z_{\mathrm{pred}}/z_{0,2})}\right]\bigg/\left[1+\frac{\ln a}{A-\ln[G/(fz_{0,2})]}\right]. \tag{10}$$

This follows from (A3), which includes details of the derivation (Appendix A).

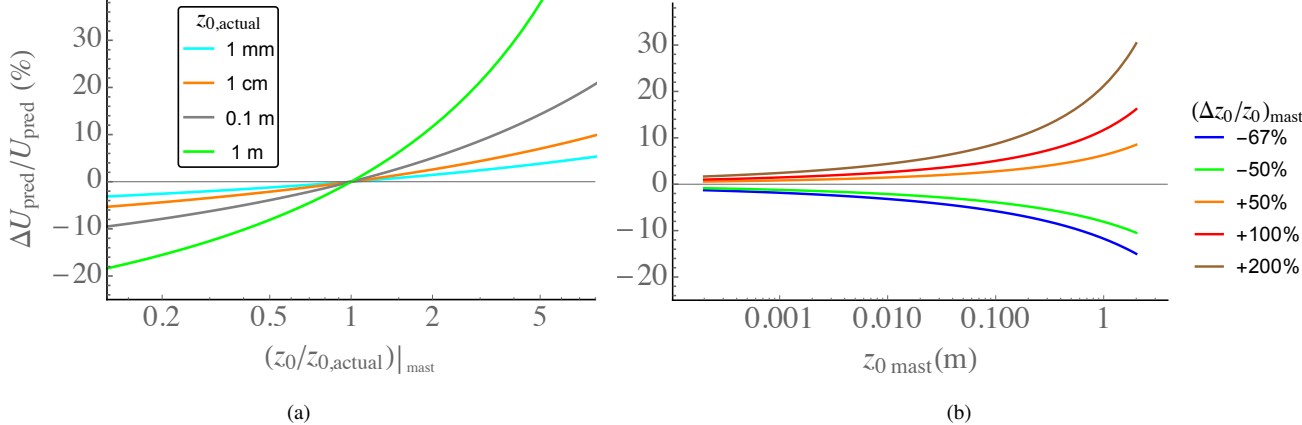

**Figure 3.** Error in predicted wind speed due to error in background roughness at measurement site via Eq. (8), for observation height $z_{\mathrm{obs}}=60\,\mathrm{m}$ and prediction (hub) height of $z_{\mathrm{pred}}=100\,\mathrm{m}$. Left (a): error versus ratio ($=a$) of estimated to actual background $z_0$. Right (b): error vs. background $z_0$ at observation mast; uncertainties of $\{-67\%,-50\%,50\%,100\%,200\%\}$ correspond to $a=\{\frac{1}{3},\frac{1}{2},1.5,2,3\}$.

The sensitivity of hub-height (predicted) wind speed to $z_{0,1}$, via (8), is shown in Figure 3 for the case of $z_{\mathrm{obs}}=60\,\mathrm{m}$ obser-
vation height and a hub height of 100 m. Similarly, the uncertainty in predicted wind speed due to uncertainty in prediction-site roughness $z_{0,2}$, via (10), is displayed in Figure 4.




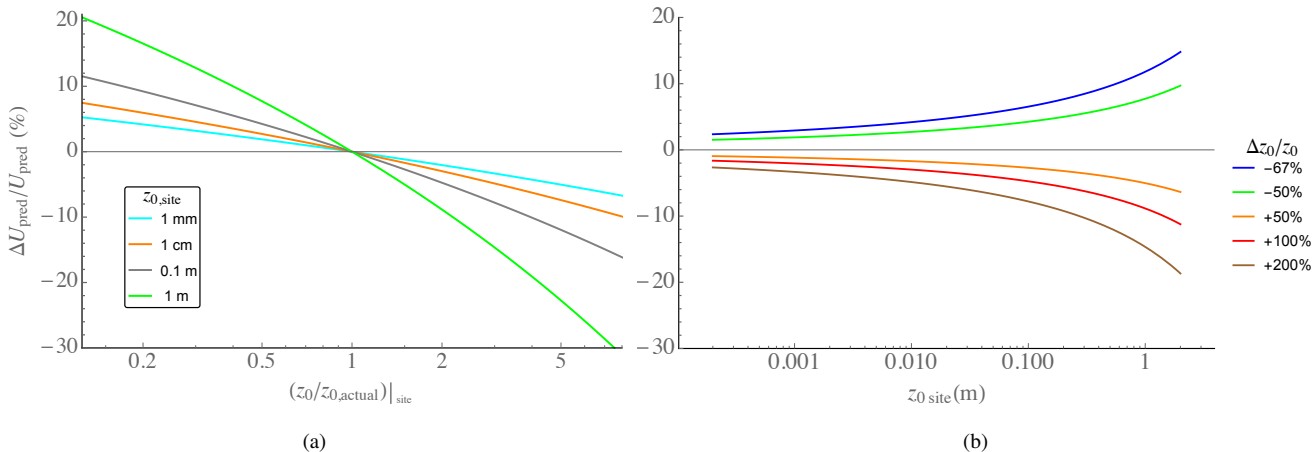

**Figure 4.** Error in predicted wind speed due to error in background roughness at prediction site via Eq. (10), for observation height $z_{obs} = 60\,\mathrm{m}$ and prediction (hub) height of $z_{pred} = 100\,\mathrm{m}$. Left (a): error versus ratio ($= a$) of estimated to actual background $z_0$. Right (b): error vs. background $z_0$ at observation mast; uncertainties of $\{-67\%, -50\%, 50\%, 100\%, 200\%\}$ correspond to $a = \{\frac{1}{3}, \frac{1}{2}, 1.5, 2, 3\}$.

The estimated relative uncertainty in predicted wind speed ($\Delta U_{pred}$) is first plotted versus fractional roughness uncertainty $a$ for a number of different measurement-site background roughnesses ($z_{0,mast}$), and then also plotted against $z_{0,mast}$ for different relative roughness uncertainty ($\Delta z_{0,mast}/z_{0,mast} = a - 1$), expressed as a percentage. For small background roughnesses one can see less effect on predicted wind speed for a given roughness-error or uncertainty, with a nearly linear dependence of

relative windspeed uncertainty upon $z_{0,mast}$ for measurements taken over smooth land or water ($z_{0,mast} < {\sim} 1\,\mathrm{cm}$). For larger magnitudes of roughness uncertainty, as expected, one sees larger expected uncertainty in wind speed as well; this effect is reduced for smooth measurement sites (in conjunction with the previous statement). Also, for higher background roughnesses, the sensitivity of wind speed to (relative) roughness error is amplified, as shown by the green lines in the left-hand ('a') plots or the right-most (high $z_0$) part of the right-hand ('b') plots of Figs. 3–4. Comparing Figure 4 to Figure 3, one also sees that

the effect of a given change (or uncertainty in) $z_{0,2}$ has the opposite sign of the corresponding effect due to an equal change in $z_{0,1}$, but with the measurement/mast location's roughness $z_{0,1}$ having a larger effect than the prediction site roughness $z_{0,2}$. That is, the magnitudes of $\Delta U_{pred}(\Delta z_{0,1})$ in Figure 3 are larger than the magnitudes of $\Delta U_{pred}(\Delta z_{0,2})$ displayed in Figure 4.

### Roughness bias and combined effect of $z_0$-sensitivities at measurement and prediction sites

Above we saw that wind speeds predicted via the GDL (3) with roughness-affected (logarithmic) wind profile (1) can be more

sensitive to $z_{0,1}$ than to $z_{0,2}$. Thus, for an overall bias in roughness estimates, we should expect a net bias in wind speed predictions via wind-atlas methods. In other words, for roughnesses that are systematically overestimated (or underestimated) by the same factor $a_{bias}$ at measurement and prediction sites, we then expect a corresponding bias in predicted mean wind speed. This effect is shown by Figure 5, which displays the fractional change in predicted wind speed as a function of fractional change in measurement and prediction-site $z_0$, for combinations of $\{z_{0,1}, z_{0,2}\}$ that span typical application (colored lines).





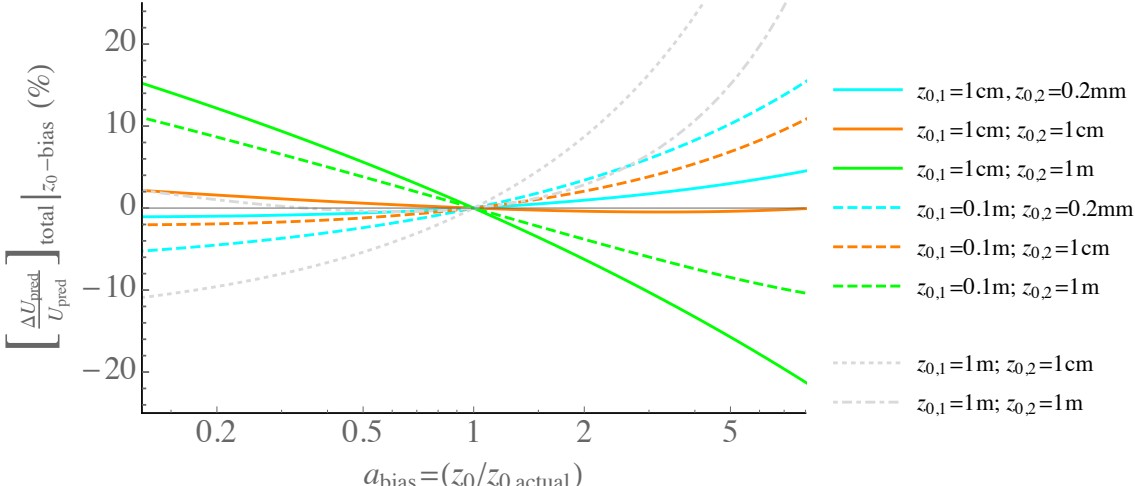

**Figure 5.** Total error in predicted wind speed due to a bias ($a_{bias}$) in background roughness at both prediction and measurement sites, for different combinations of background roughness at the sites. As in Figs. 3–4, observation height is $z_{obs} = 60$ m and prediction (hub) height is $z_{pred} = 100$ m.

As one might expect, for measurement and observation sites having similar background roughness, the change $\Delta U_{pred}/U_{pred}$ is relatively small, especially for systematically underestimated roughness lengths ($a_{bias} < 1$). Figure 5 also shows that for small biases ($a_{bias} \to 1$), the wind speed prediction error is larger when the roughnesses at measurement and prediction sites are dissimilar. However, for roughness errors of a factor of $\sim 2$ or more, the nonlinearity of (3) with (1) complicates the dependence

of $\Delta U_{pred}$ on $a_{bias}$. In addition to the typical range of $z_0$ used in wind resource estimation (colored lines) Figure 5 also shows the gross effect of measurement over forest (or effectively more complex terrain, i.e. with effective roughness $z_{0,1}=1$ m, denoted by grey lines); one can see the corresponding increase in $\Delta U_{pred}/U_{pred}$ for such cases when there is overestimation of $z_0$—even if $z_{0,2}$ and $z_{0,1}$ are both 1 m.

In contrast to a possible bias in roughness assignment, one can imagine a 'worst case' scenario having a negative error in

$z_{0,1}$ and positive error in $z_{0,2}$ (or vice-versa), e.g. $a_1 = 1/a_2$. In this scenario the result resembles the plots in Figure 5, but rotated 45° with the y-axis stretched by a factor of 2: cases with $z_{0,1} = z_{0,2}$ no longer have small error, but all the lines show a large uncertainty for $a$ far from 1 (e.g. $\pm 40\%$ at $a_1^{\pm 1} = a_2^{\mp 1} = 0.1$ for $z_{0,1}=1$ cm, $z_{0,2}=1$ m, corresponding to solid green line), and all lines have $\Delta U_{pred}=0$ for $a = 1$.

A more general situation is that of independent errors in roughness assignment at different sites. In this limit, one forsees

a distribution of $\Delta U_{pred}$, given uncertainties in $z_{0,2}$ and $z_{0,1}$ (basically $P(z_{0,2})$ and $P(z_{0,1})$). Two examples of this are given in Figure 6. The figure shows $P(\Delta U_{pred}/U_{pred})$ for the cases of winds observed over grass but predicting winds over grass or forest, where the grass and forest $z_0$ have log-normal distributions $P(a)$ with means and widths given for the combined samples in Table 1. Following the earlier examples, the observation height is taken as 60 m and prediction (hub) height is 100 m. In the figure one can see the combined effect of different roughness distributions and uncertainties, particularly for the case of grass-





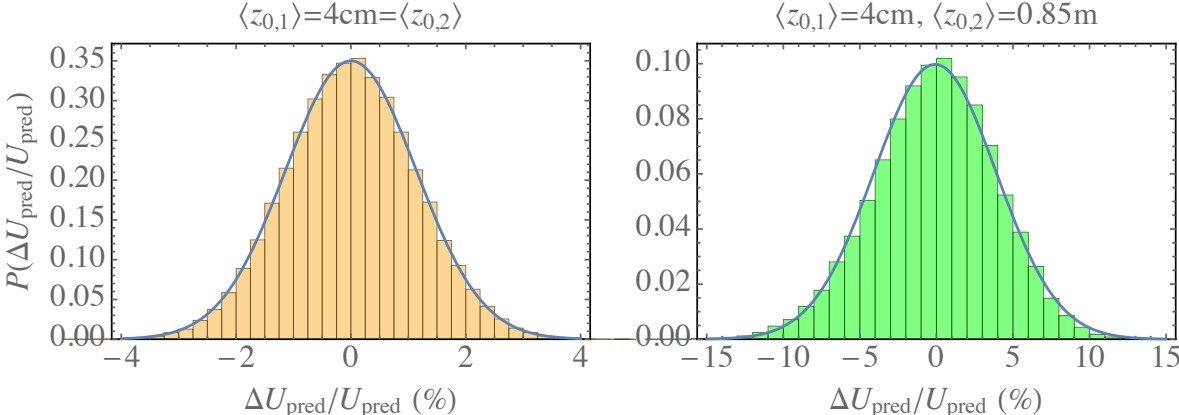

**Figure 6.** Distribution of error in predicted wind speed, given distributions $P(z_{0,2}/\langle z_{0,2}\rangle)$ and $P(z_{0,1}/\langle z_{0,1}\rangle)$ at prediction and measurement sites. Left: prediction from grass to grass; right: from grass to forest. Input $P(z_0)$ follow from elicited samples in Table 1; see text. Blue line is normal distribution based on calculated mean and standard deviation. As in Figs. 3–5, $z_{obs} = 60$ m and $z_{pred} = 100$ m.

to-forest (right plot in Fig. 6). For this case the half-width of the grass $P(z_{0,1})$ corresponds to 117% (where $\langle z_{0,1}\rangle = 4$ cm) and that for the forest corresponds to 141% of $\langle z_{0,2}\rangle = 0.85$ m following Table 1. The combined effect gives wider error distributions $P(\Delta U_{pred})$ for the grass-to-forest case than for the grass-grass case, as expected from e.g. Figure 5; the standard deviations corresponding to the $z_0$-induced mean-wind error distributions in Figure 6 are 1% and 4% for the predictions over
5 grass and forest, respectively (and both error distributions are nearly Gaussian, with skewnesses of 0.02 and −0.2). To be yet more conservative, if we follow section 2.1.2 using a gross estimate of observational $z_0$-uncertainty equivalent to a half-width (roughness uncertainty factor) of $w_{\langle z_0\rangle_{RS}} \sim 3$, the uncertainty $\sigma_{\Delta U_{pred}/U_{pred}}$ (distribution widths) for the two cases shown in Figure 6 grow to 8.6% and 14%, respectively. Towards practical consideration for wind engineers, we also point out that for prediction over water (again from $z_{obs}=60$ m to $z_{pred} = 100$ m with $z_{0,1}=4.1$ cm), again using the conservative roughness-
10 uncertainty estimate $a = w_{\langle z_0\rangle_{RS}} \sim 3$ leads to uncertainty in $U_{pred}$ that exceeds 6% and an error distribution that is somewhat non-Gaussian (skewness≈0.6, plot not shown); we provide this number to demonstrate the roughness-induced uncertainty expected when using land-based measurements for off-shore predictions.

### 3.2.2 Sensitivity of predicted energy production to background $z_0$

The uncertainty in background roughness can also be translated into AEP uncertainty, by employing a relation between wind
speed and AEP—i.e. via a turbine (or perhaps windfarm) power curve. The propagation of $z_0$-uncertainty to AEP follows that derived for wind speed above, but with some assumptions. First, we assume Weibull-distributed winds, which is standard practice in wind energy, and also facilitates analytical derivation of a bulk relation between AEP and mean wind speed $\langle U\rangle$. Because power curves in practice do not have a 'kink' at rated wind speed, but rather a smooth transition from the ideal $\langle U\rangle^3$-regime to the maximum (rated) power regime of operation (e.g. Wagner et al., 2011), we can derive an analytical effective power-curve




form, expressible as a function of $\langle U \rangle / V_{\mathrm{rat}}$, i.e. Eqn. B1 (shown in Appendix B). To accomplish analytical integration and readily relate mean wind speed (or Weibull-$A$ parameter) per turbine rated speed, some mathematical approximations are used, wherein we also assume that the Weibull-$k$ parameter is close to a value of 2 (within 10-20%). The analytical power-curve form $P(U/V_{\mathrm{rat}})$ then leads to a power-law relation between normalized AEP and wind speed: AEP $\propto (\langle U \rangle / V_{\mathrm{rat}})^p$, where the

power-law exponent $p$ is also a function of $\langle U \rangle / V_{\mathrm{rat}}$, as shown in Appendix B.

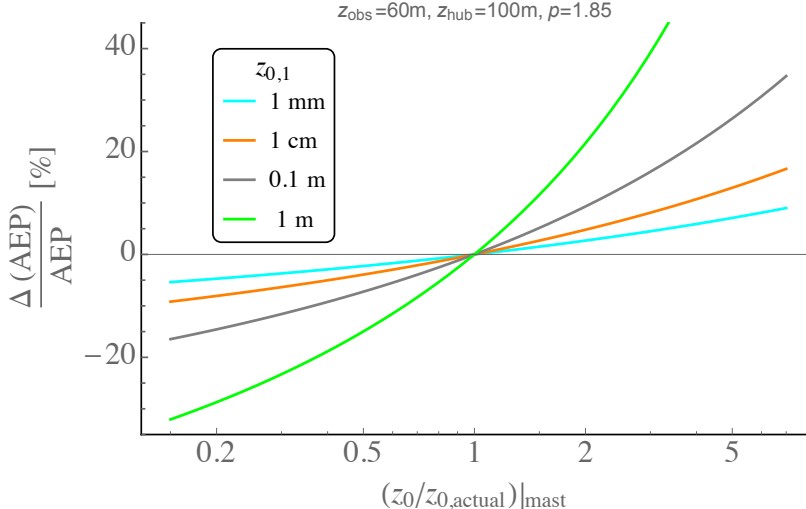

**Figure 7.** Sensitivity of (error in) predicted normalized power due solely to error in background roughness at measurement site, versus ratio of estimated to actual background $z_0$ at observation mast (i.e. 1+relative error, Eq. 9), for various values of actual $z_{0,1}$. Observation height is 60 m and hub height is 100 m, as in Figs. 3–6.

Figure 7 shows an example of AEP sensitivity to fractional roughness uncertainty of the observation site (ratio of estimated to actual $z_{0,1}$, i.e. $a = \Delta z_{0,1}/z_{0,1} + 1$ as in Eqn. 9), for the case of $\langle U \rangle = 0.7 V_{\mathrm{rat}}$; the latter translates to a power exponent of $p \simeq 1.85$ for the analytical power curve form elucidated in Appendix B (c.f. Fig.12b). For Figure 7 we consider the same situation as used for Figs. 3–5 ($z_{\mathrm{obs}}$=60 m and $z_{\mathrm{pred}}$=100 m). As one might expect, the AEP uncertainty—due to uncertainties in

$z_{0,1}$, $z_{0,2}$ or their combined effect with a common bias—simply resembles the wind speed uncertainty plots shown in Figures 3–5: the vertical axis of the plot appears stretched by a factor of $p$ ($\simeq 1.85$). An analogous plot of the distribution of AEP error follows similarly; for a given value of $p$ (here 1.85) the horizontal ($x$-) axes in the plots of Figure 6 are stretched by a factor of $p$ to give the distribution of $\Delta$AEP.

### 3.3 Effect of uncertainty in background roughness upon wind resource predictions

In order to give examples (and realistic numbers) useful to wind engineers, in this section we translate the observation-based (sec. 3.1.1) and user-based (sec. 3.1.2) roughness uncertainties into uncertainties of predicted mean wind speed and AEP, for the observation and user-survey examples treated in sections 3.1.1 and 3.1.2, respectively.





### 3.3.1 Uncertainty in predicted mean wind speeds

The relative uncertainties implied by roughness lengths calculated via surface-layer wind speed measurements were outlined in section 3.1.1, for the seemingly ideal grassy terrain east of Høvsøre. The half-widths of the roughness distributions for the homogeneous sectors were found to be on the order of a factor of 3 times $\langle z_0 \rangle$, while the uncertainty in obtaining a mean (representative) roughness was found through bootstrap-resampling to be much smaller, about 5%; this result came whether $z_0$ was calculated from speeds at multiple heights in the ASL or from sonic anemometer measurements of $U$ and $u_*$ in the ASL. However, despite similar distribution widths and similar *apparent* uncertainty in mean-estimation, the $\langle z_0 \rangle$ themselves differed by roughly one-half order of magnitude, i.e. a factor of $\sim$3 when determined in these two different ways. Thus we first consider (conservatively) a relative uncertainty of $a \sim 3^{\pm 1}$ for $z_0$, for the typical resource-assesment heights ($z_{\mathrm{obs}}$=60 m, $z_{\mathrm{pred}}$=100 m) used in Figs. 3–7. As seen in Figure 5, for systematic (bias) overestimates of $a \equiv \Delta z_0 / z_0$ and a mean roughness length at the observation site of 1 cm, this translates into wind-speed uncertainty values of less than 1% when predicting 100-m winds over the same roughness, and gives $\Delta U_{\mathrm{pred}}$ of $\{2\%, -2\%, -6\%, -10\%\}$ for predictions over roughnesses of $z_{0,2} = \{0.2\,\mathrm{mm}, 3\,\mathrm{cm}, 30\,\mathrm{cm}, 1\,\mathrm{m}\}$. For the same magnitude of systematic underestimate ($a \sim 1/3$) the corresponding $\Delta U_{\mathrm{pred}}/U_{\mathrm{pred}}$ are $\{-1\%, 2\%, 5\%, 9\%\}$ for these $z_{0,2}$, and an uncertainty of 1% for 100-m winds predicted over the same roughness as the measurement site. Thus we see about 1% uncertainty in $U_{\mathrm{pred}}$ for these typical heights and the same observation/prediction roughness, while using such observations to predict winds over e.g. nearby forested land incur higher uncertainties, with magnitudes of 5–10%—without yet considering modeling the flow over such. To get estimates of $\Delta U(\Delta z_0)$ for other observation/prediction heights and roughnesses, we remind the reader that these can be obtained from (8)–(10).

For the uncertainties inherent in user-provided roughness lengths, we address the two cases treated in section 3.1.2. The grass case is similar to that considered in the Høvsøre analysis above, with a mean roughness of about 4 cm. If we take the half-width of the expected user-input distribution of $z_0$, i.e. $\exp\{\sigma_{\ln[z_0/\langle z_0 \rangle_g]}\}$ from Table 1, then we can again arrive at estimates for the wind-speed uncertainty (this is also a bit conservative, because it gives larger uncertainties than the bootstrap-derived half-width). Again assuming 'typical' application heights ($z_{\mathrm{obs}}$=60 m, $z_{\mathrm{pred}}$=100 m), for predictions over site roughnesses $z_{0,2} = \{0.2\,\mathrm{mm}, 1\,\mathrm{cm}, 30\,\mathrm{cm}, 1\,\mathrm{m}\}$ and a $z_0$-bias of $2.2^{\pm 1}$ ($\pm 120\%$ from Table 1) we obtain $U_{\mathrm{pred}}$ uncertainties of $+\{3\%, 1\%, -3\%, -6\%\}$ and $-\{2\%, 0.4\%, -3\%, -5\%\}$ respectively. These roughly correspond to (a proxy of) the industry-wide uncertainty in predicted wind speeds (with this $z_{\mathrm{obs}}, z_{\mathrm{pred}}$) for observations over a background roughness like the grass in Figure 2. For the surveyed 'forest' roughness in that figure, we get corresponding $\Delta U_{\mathrm{pred}}$ following Table 1 for the case of all-site biases ($\pm 141\% \rightarrow a \approx 2.4^{\pm 1}$ applied to both $z_{0,1}$ and $z_{0,2}$). For predictions from observations over such a site, applied to turbine sites having $z_{0,2}$=$\{1\,\mathrm{cm}, 10\,\mathrm{cm}, 1\,\mathrm{m}\}$ we get $\Delta U_{\mathrm{pred}} \approx (+)\{11\%, 9\%, 3\%\}$ for systematic overestimates and $(-)\{6\%, 4\%, 0.3\%\}$ for systematic $z_0$-underestimation. The latter finding is rather significant, as it implies that an underestimation of forest roughness lengths is safer than overestimating $z_0$ when using EWA-based methods for wind resource estimates (e.g. WAsP and similar). This is consistent with common practice: while recent evidence from e.g. direct LIDAR scans of forests suggests $z_0$ should be at least several meters there (Boudreault et al., 2015), industrial practice has been to use $z_0$ of 1 m or less (e.g. Troen and Petersen, 1989; Mortensen et al., 2014).





### 3.3.2 Uncertainty in predicted energy production

The magnitude of $z_0$-induced AEP uncertainty for typical simple sites over land (i.e. $z_{0,1} \sim 1$–$10\,\mathrm{cm}$) depends in general on the ratio of $\langle U \rangle / V_{\mathrm{rat}}$ (for 'classically-behaved' turbines), because the relationship between $\langle U \rangle$ and AEP depends on such—most simply expressed via the exponent

$$p = \frac{\ln(\mathrm{AEP})}{\ln\langle U \rangle} \qquad \text{for a power-law relation} \qquad \mathrm{AEP} = \langle U \rangle^p, \qquad (11)$$

detailed in Appendix B. As mentioned in the previous section, with regards to uncertainty in the background roughness of either the observation or prediction site (or for a bias across both sites), the sensitivity plots of $\Delta\langle U \rangle$ per given roughness errors are simply translated into analagous AEP-sensitivity figures via stretching the vertical axes by a factor $p$ (as was done to get Figure 7 from Fig. 3a); similarly the horizontal ($\Delta U$) axis in Figure 6 is stretched by a factor $p$. Since $p$ basically varies
between $\sim 0.8$ and 2.5 (over the reasonable range of $\langle U \rangle / V_{\mathrm{rat}} \sim 0.5$–$0.9$), then the mean wind speed uncertainties quoted in the previous subsection can be simply multiplied by a factor of $\sim 0.8$–$2.5$, depending on the expected turbine power curve and subsequent $p$.

For most general practical use, we ultimately consider roughness error distributions and the consequent AEP error distributions, such as those shown in Figure 6. For independent roughness error distributions at measurement and prediction sites,
and assuming log-normal distributed $\Delta z_0$ (as demonstrated in subsections 3.1.1 and 3.1.2 for measured and user-estimated distributions), via (8) and (10) we can obtain distributions of $\Delta\mathrm{AEP}$. The uncertainty in $z_0$ can be expressed in terms of the dimensionless width $w_{z_0} / \langle z_0 \rangle$; for a given width we can synthesize distributions of $z_{0,1}$ and $z_{0,2}$, and then find the standard deviation of the resulting distribution of $\Delta\mathrm{AEP}$. We do such Monte Carlo simulations over the range of dimensionless widths from 5% to 500%, for the same $\{z_{0,1}, z_{0,2}\}$ pairs and observation/prediction heights as used in Fig. 5; the results are shown in
Figure 8.

Figure 8 shows AEP uncertainty versus roughness uncertainty; the latter is expressed as the dimensionless width $w_{z_0} / \langle z_0 \rangle$ of the $z_0$-distribution, calculated via the standard deviation of $\ln(z_0/\langle z_0 \rangle)$ as in Eq. 4. Following the previous sub-section's analysis (where $z_{\mathrm{obs}}=60\,\mathrm{m}$, $z_{\mathrm{pred}}=100\,\mathrm{m}$ and $p = 1.85$), Fig. 8 shows that for actual measurement-site roughness $z_{0,1} \approx 1$–$10\,\mathrm{cm}$, given a relative $z_0$-uncertainty of 100% (corresponding to $w_{z_0} \sim z_0$ as in subsection 3.1.2), the GDL/$z_0$-induced AEP uncer-
tainty ranges from $\sim 5\%$ (for $\langle z_{0,1} \rangle=1\,\mathrm{cm}$, predicting over water) to 15% (for $\langle z_{0,1} \rangle=10\,\mathrm{cm}$, $\langle z_{0,2} \rangle=1\,\mathrm{m}$). For the statistical uncertainty example of (mostly) homogeneous flat farm/grassland shown previously in Fig. 1 (section 2.1.2), taking the the relative background roughness uncertainty factor to be equivalent to the width of the $\ln(z_0)$-distribution (centered around $\sim 1.4\,\mathrm{cm}$ for wind directions from $\sim 45$–$120°$), i.e. $a \sim 3^{\pm 1}$, leads to a similar AEP uncertainty range, roughly 6–16% for predicion sites ranging from water to forest/urban. However, such an uncertainty estimate seems large, and may be explained considering
Table 2. For industrial use, wind engineers (e.g. in medium or large companies) in effect assign a kind of ensemble-average roughness length for any given land-use type; considering e.g. the case of taking three 'community-accepted' values for the grass site as in Table 2, i.e. a relative $z_0$ uncertainty of roughly 50%, then from Figure 8 one sees that the AEP uncertainty drops to 4–10%. We remind that these AEP uncertainty values correspond to the case of observation and prediction heights of 60 m and 100 m, respectively: the slight dependence of $\Delta\mathrm{AEP}$ on $z_{\mathrm{obs}}$ and $z_{\mathrm{pred}}$ modifies the uncertainty for other heights.



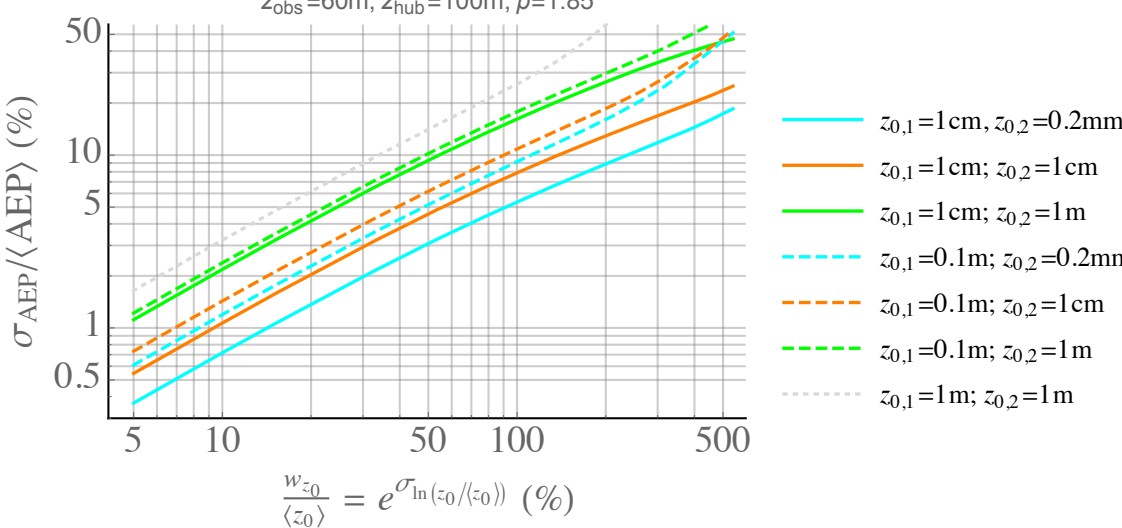

**Figure 8.** Uncertainty in AEP versus (relative) roughness uncertainty, due to the combined effect of observation/prediction site roughness-uncertainty; independent log-normal roughness distributions assumed, with dimensionless width $w_{z_0}/\langle z_0 \rangle$. Standard deviation of AEP shown for different combinations of $\langle z_{0,1} \rangle$, $\langle z_{0,2} \rangle$. Case shown for $p = 1.85$, i.e. AEP$= U^{1.85}$ as in Fig. 7.

Aside from the weak dependence on measurement/prediction heights, one also sees a basic power-law form emerging for the AEP estimates:

$$\frac{\sigma_{AEP}}{\langle AEP \rangle} \propto \sim \frac{w_{z_0}}{\langle z_0 \rangle}, \tag{12}$$

particularly for relative roughness uncertainties (i.e. widths of the distribution $P(\ln z_0 / \ln \langle z_0 \rangle)$) that are $w_{z_0}/\langle z_0 \rangle < \sim 1\text{–}2$.

5  We also note again that we have focused here on the AEP uncertainty caused by uncertainty in background roughness, rather than the $z_0$ uncertainty itself. Further details of the latter are the subject of ongoing work and another paper, and here we point to Figure 8 as the significant result: for a given uncertainty in $z_0$, one can find the corresponding uncertainty in AEP due to use of the GDL/EWA method.

## 4 Conclusions

10 We remind first of the context of this work, i.e. the 'European Wind Atlas' (EWA) method (Troen and Petersen, 1989),[5] which employs the geostrophic drag law (3): this allows for so-called horizontal extrapolation, whereby mean wind speed measured at a site with one background roughness can be used for predicting the mean wind at another site having potentially different surface characteristics, assuming the sites are forced by the same pressure gradient (geostrophic wind). For separated

---

[5]The EWA method is implemented in 'WAsP' and related software (e.g. 'WindPRO', 'WindFarmer', etc.).





measurement/prediction sites where the GDL is valid,[6] resource assessments that include $z_0$ tend to be better than assessments that ignore it (such as those based only on observed shear exponent, c.f. Kelly, 2016), especially when the sites are in terrain with different background roughness; consequently the EWA method has been used in wind energy for decades. The need for and justifaction of this method is also implied by Figure 4, which displays the sensitivity of EWA-predicted winds to prediction-site roughness ($z_{0,2}$); it can thus be used also to show how much the predicted mean wind changes due to $z_{0,2}$ differing from the measurement-site roughness $z_{0,1}$. For ratios $z_{0,2}/z_{0,1}$ deviating significantly from 1 (i.e. taking the $x$-axis of Fig. 4a as this ratio), depending on the roughnesses involved, a significant $\Delta U_{\mathrm{pred}}$ can result—and the EWA method is needed to account for such. One can see that if $z_{0,2}$ differs from $z_{0,1}$ by a factor of 5, the predicted mean wind may be affected by ∼5–25%; subsequently the AEP could change by a factor of up to 2.5 times this.

Use of the EWA method is affected by uncertainty in the background roughness at measurement and prediction sites, which can be significant; this leads to uncertainty in resource predictions as shown in section 3. Both user-implicit (§3.1.2) and definition-related (§3.1.1) uncertainties in roughness length are found to effectively be (treatable as) roughly of the same order of magnitude, and they lead to an uncertainty in prediction of mean wind speed and AEP when the EWA method is used to 'horizontally extrapolate' measurements from one location and background roughness to another. The uncertainty in prediction is slightly more sensitive to measurement-site roughness $z_{0,1}$ than prediction-site roughness $z_{0,2}$, as seen in equations 8–10 and displayed in e.g. Figs. 3–4. However, there is also a minor dependence on measurement and prediction heights via the vertical wind profile used within the EWA method (log-law implicit in Eqs. 8, 10), shown by Fig. 10 in Appendix A.

As mentioned in section 3.1.1, even in ideal (steady, neutral) conditions, the mean roughness $\langle z_0 \rangle$ obtained from observations and (1) via different calculation methods in the surface layer, such as using wind speeds at multiple heights or alternately wind speed with friction velocity, differs by an amount that appears to greatly exceed the uncertainty derived for any given method. For example, boot-strapped distributions of $\langle z_0 \rangle$ for the homogeneous flat grassland sectors at Høvsøre had relative widths (approximate uncertainty) well under 10% when using (1) and $U_{\mathrm{obs}}$ in the surface layer, whether calculated with or without $u_*$; but the ratio of the means from the corresponding boot-strapped distributions was roughly 3. In contrast, the uncertainty of $z_0$ estimated from polls of two groups of wind resource assessment experts (for grassland and forest) in section 3.1.2 was on the order of $z_0$ itself, i.e. $w/\langle z_0 \rangle \sim 1$ when estimated from single values of $z_0$ as in Table 1; it is however smaller if assuming that wind engineers may gauge roughness from multiple accepted sources, as in the example of Table 2.

We note that more exact quantification of 'measured' roughness uncertainty involves consideration of numerous other factors, from ABL physics and fluid dynamics to inhomogeneous boundary conditions and turbulent transport. Likewise, more accurate characterization of epistemic user-based (industry-wide) uncertainty would likely require a much wider survey, for a greater number of roughnesses. Here we have made a basic evaluation of the main roughness uncertainty components and their approximate magnitudes—focusing first on what resultant uncertainty can be expected in a wind resource prediction, given some level of roughness uncertainty. The latter focus leads to analysis culminating in Figure 8, which visualizes a primary

---

[6]The GDL applies to sites having approximately the same latitude and geostrophic-scale forcing (roughly the distribution of geostrophic wind); the scale of spatial variations in the geostrophic wind depends on the terrain complexity, and can vary from a several tens of kilometers in simple terrain down to just a few kilometers in very complex terrain or near coasts; c.f. Troen et al. (2014), Hahmann et al. (2015).





result of this work: the uncertainty in AEP (or scaled mean wind) predicted via the EWA method, for a given uncertainty in background roughness length and pair of surface types (roughnesses) at separate prediction and measurement sites. From Figure 8 we see the basic trend for uncertainty in mean wind speed or AEP behaves as approximately $(w/\langle z_0 \rangle)^{6/7}$ in the dimensionless roughness uncertainty regime $w/\langle z_0 \rangle < \sim 200\%$, i.e. just within the range we have estimated.

There are other sources of uncertainty implicit in use of the EWA method, in addition to the roughness lengths. Additional
uncertainties include the applicability of the GDL (see footnote 6), the the constants $(A, B)$ within (3), and the actual form and/or use of (3) with arguments averaged in an ensemble (or spatial) sense. These are beyond the scope of the current paper. However, as for applicability of the GDL, regarding the distance between measurement and prediction sites, we remind the reader that (fine-resolution) mesoscale models give an indication of the spatial extent (and direction) of variations in the geostrophic wind, and refer the reader to e.g. Hahmann et al. (2015) and Troen and Petersen (1989). As to the distance over
which one may 'horizontally extrapolate' in more complex terrain, this depends upon the observation and prediction heights along with the terrain complexity (as e.g. ruggedness index ('RIX') or local elevation variability Kelly et al., 2014a; Kelly, 2016); we point the reader to Clerc et al. (2012) and Troen et al. (2014). The (minor) uncertainties due to GDL constants $(A, B)$ are the subject of ongoing work (e.g. Floors et al., 2015), and the averaging issue is currently seen to be secondary due to the well-behaved nature of (8) and (10) and the magnitude of $z_0$ variations expected.

Additional uncertainties can also arise due to the use of a (mean) wind profile expression, such as the simple "log-law" (1) invoked here. One uncertainty is due to the applicability of a given profile model. Following Troen and Petersen (1989) and due to the the statistical dominance of neutral conditions (Kelly and Gryning, 2010), we have used the (surface-layer) form (1) applicable in neutral conditions; further, we limit our observational analysis to neutral steady conditions, and observations to be within the surface-layer, where the logarithmic profile is valid and the roughness length is simply defined. However, deviations
from logarithmic may occur above the surface layer, such as for the prediction height considered in the figures (100 m) for a small fraction of ABL depths (Liu and Liang, 2010) occuring in reality (i.e. depths less than $\sim 2z_{\mathrm{pred}}$ (Pedersen et al., 2014) or 200 m in this case). This ABL-depth effect is negligible for $z_{\mathrm{pred}}$ close to $z_{\mathrm{obs}}$ (and $z_{0,1}/z_{0,2}$ near 1), and is minor for the heights considered. However, an additional uncertainty dependent upon the ABL depth could be modelelled following Kelly and Gryning (2010) and Liu and Liang (2010), or alternately a better profile form (e.g. Kelly and Gryning, 2010) could be
invoked along with the GDL, particularly to reduce uncertainties for predictions well above 100 m or in areas where lower-level jets are expected. Another uncertainty arising implicitly from the profile model, as analyzed here, is due to considering the same $z_0$ for use in both the profile model and the GDL. That is, the wind profile reacts to a more *local* roughness, whereas the GDL reacts to a geostrophic-scale $z_0$. In Troen and Petersen (1989) the latter is obtained by taking a weighted geometric average of $z_0$, where $\ln z_0$ is integrated radially (in space) upwind from a given location but with a weighting function that
emphasizes $z_0$ at the site and decays with distance upwind;[7] thus the local and geostrophic $z_0$ can differ slightly. This is not likely to have a major effect in the analysis here, since the Hovsore sectors considered were ideal and without significant homogeneity, such that the upwind-averaged roughness is within 10% of the local $z_0$. However, it is worth noting that for large

---

[7]The EWA roughness-averaging weighting function is prescribed as $\exp(-r/\ell_r)$, where $r$ is the distance upwind, $\ell_r$ is a length scale generally taken to be 10 km (as e.g. default WAsP value), and the integration is carried out to 20–30 km (roughly half the Rossby radius).



roughness-changes within $\sim 10\,\text{km}$ of a site, the geostrophic $z_0$ will differ from the site's $z_0$; equations 8–10 can be re-cast for such. The effect on roughness-uncertainty incurred through such spatial averaging is expected to be smaller than the rough factor of 3 (200%) found and presented above, though this systematic evaluation of this effect is still a subject of ongoing research. Analogously, the height-dependent effect of inhomogeneities upon roughness (i.e. above the ASL)—in particular its uncertainty—is also under study, but is also expected to be minor for simple terrain.

Vertical extrapolation has not been treated explicitly here, though it is implicit in the vertical profile used to estimate $u_*$ from observed wind for use in the GDL. Such treatment, in conjunction with taking the 'profile roughness' and geostrophic-scale roughness to be the same, is a choice that we have made to facilitate systematic modeling of roughness-induced uncertainty; thus we have been able to estimate the effect of roughness, which occurs through both the wind profile ('vertical extrapolation') and through invocation of the GDL ('horizontal extrapolation'). A separate model for the uncertainty in vertical extrapolation

using a logarithmic-based profile (as in the EWA/popular wind software, e.g. WAsP), but without considering roughness uncertainty, is given in Kelly and Troen (2016) and Kelly (2016). Treating the $z_0$-related uncertainties separately, per the geostrophic drag law and wind profile, is the subject of continuing work beyond the scope of the current article.

### 4.1    Applications and implications

In increasingly complex terrain the actual surface roughness becomes less significant compared to terrain slope, with regards

to affecting the flow. However for horizontal extrapolation, the aggregate effect of the (complex) terrain-induced drag leads to an increase in the effective geostrophic-scale roughness (Beljaars et al., 2004; Kelly et al., 2014a). Thus the geostrophic-drag and roughness uncertainty analysis given in this work can also be applied towards improved use of microscale models in complex terrain, when horizontal extrapolation is involved. In particular, computational fluid dynamics solvers (e.g. RANS and LES), when employed using different simulation domains for measurement and windfarm sites, are typically used to calculate

terrain-induced flow perturbations ('speed-up' factors) at the respective sites. But for domains having different degrees of complexity (or potentially different resolutions)—and thus different large-scale drag—then the use of the geostrophic drag law (or any analogous empirical algorithm/method) demands that measured wind statistics must additionally be transformed properly, accounting for differences in the effective domain-scale mean roughness in the two domains (per wind direction). So uncertainty in characterizing the effective roughness due to terrain drag can be translated into a corresponding uncertainty

in mean wind (or AEP) via the framework presented here. Alternately, for a given pair of (observation, prediction) sites, the uncertainty in mean wind prediction due to neglect of terrain drag can be estimated: a bias is introduced, whereby the effective geostrophic roughness is underestimated. From Figure 5 one can see for example that, for sites having the same effective roughness ('complexity') of $z_{0,\text{eff}} \sim 1\,\text{m}$ and with an underestimation of one order of magnitude ($a_{\text{bias}} \simeq 0.1$), a positive error $\Delta U_{\text{pred}} \sim 2\%$ is incurred.

Another implication of this work applies to assessment in forested regions. Some work on characterizing profile-amenable roughness over forest (e.g. Bosveld, 1997; Tian et al., 2011; Boudreault et al., 2015) implies that $z_0$ over forest is larger than what has been typically assigned in wind resource assessment (i.e. $z_0 > 1$, not $z_0 \lesssim 1$), despite such underestimates being used for decades in the wind industry (Troen and Petersen, 1989; Mortensen et al., 2001; Emeis, 2013; Landberg, 2016). We now





see an explanation for this, looking at Figure 5: systematic underestimation leads to smaller errors in wind speeds predicted via the EWA method, compared to a positive bias on $z_0$, particularly for typical application where both measurement and turbine sites are in high-roughness areas (dash-dot line in Fig. 5) such as forest.

The roughness sensitivity/uncertainty analysis delveloped here also has application to—and implications on—the treatment of mesoscale model output for use in microscale wind flow models. In so-called meso-to-microscale 'downscaling' or wind climate 'generalization' (Hahmann et al., 2013; Badger et al., 2014), mesoscale wind output (or statistics of such) is treated in order to avoid "double-counting" of local surface-induced effects by the microscale model that have already been included in the mesoscale modelling. Additionally, the meso-micro downscaling procedure facilitates driving of the microscale flow simulation with mean winds that are appropriate as per the roughness input to both the microscale and mesoscale models, i.e. an effective geostrophic wind via the EWA method. Since any given planetary-boundary layer (PBL) scheme in a mesoscale model can react differently, for a given model resolution, it may be necessary to scale input roughnesses used in the generalization procedure. For (homogeneous ideal) output wind profiles from a particular PBL scheme and resolution, the ratio of profile-implied $z_0$ to input $z_0$ can be used with the analytic sensitivity relations developed herein, to systematically adjust the input roughness map and/or to scale the wind inputs to microscale models.

An additional application following from the roughness analysis herein—and consequently ongoing research—involves a limitation inherent in using a single characteristic (mean) roughness length. Due to the statistical nature of roughness and the significant width of measured roughness distributions (e.g. Fig. 1), an improvement would be to use $P(z_0)$ instead of mean $z_0$ in wind assessment and atmospheric flow modeling, following the suggestion of Kelly and Gryning (2010). This becomes yet more significant (and complicated) considering that the width of $P(z_0)$ tends to depend on direction and vary from site to site, and it also involves correlations with other e.g. stability (Zilitinkevich et al., 2008). Given the limited applicability of the EWA method to time series (the GDL was not explicitly derived in a statistical mean sense), refined wind resource estimates—which are essentially statistical atmospheric fluid mechanics—using (joint) distributions of roughness and stability offer potential improvement over current mean methods and are a subject of continued study.

One final application follows from the analytical form introduced here to approximate common production power curves, in a general/universal way under the assumption of Weibull-distributed wind speeds. From this, the exponent in the power-law expression relating annual energy production and mean wind speed was derived, allowing us to relate uncertainty in roughness length to uncertainty in AEP. More flexible power-curve forms can also be made from logistic functions (e.g. generalizing those of Villanueva and Feijoo, 2016) as well. Regardless of the exact form, such analytical treatment also facilitates quick computation of power for a given set of Weibull parameters, applicable to large datasets such as the Global Wind Atlas (Badger et al., 2015). Lastly we re-iterate that issues in the definition of roughness length, and specific limits of its validity, are beyond the scope of this article. However, current ongoing work includes closer examination of the (turbulent) mechanisms involved in the 'observation' of roughness length from wind measurements and heterogeneity; subsequent links to refined uncertainty characterization may follow such investigation.





## Appendix A: Geostrophic-roughness sensitivity relations: analytical forms and simplification

Here we elucidate the relations and approximations which allow translation of the partial derivatives of hub-height wind speed with regard to roughness (i.e. Eqn. 6) into sensitivity and uncertainty relations such as (8).

### A1 Sensitivity to measurement-site roughness $z_{0,1}$

First we approximate (6) by a modified power-law form that accounts for the strongest dependences ($z_{0,1}$ and $z_{obs}$), which we find to be

$$\frac{\partial \ln U_{pred}}{\partial \ln z_{0,1}} \simeq 1.1 \frac{[(z_{obs}/z_{0,1})(1 + z_{obs}/80\,\mathrm{m})]^{-1/7}}{\ln(z_{obs}/z_{0,1})}. \tag{A1}$$

This approximation is shown by the dotted lines in Figure 9 below, which also shows that it closely matches (6). Because

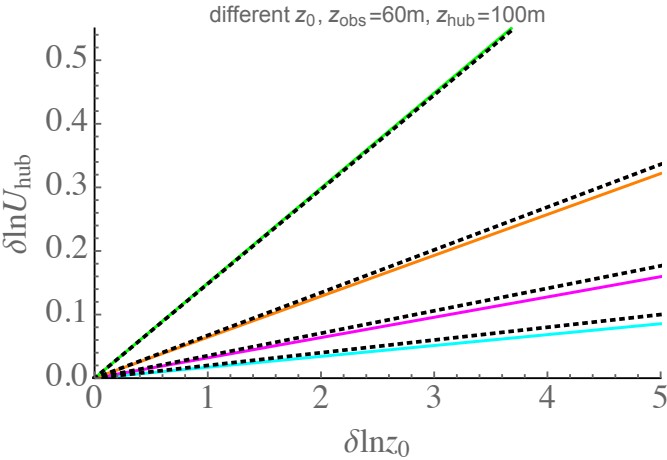

**Figure 9.** Equation 6 (solid) and its approximation, Eq. A1 (dotted), for different (correctly observed) background roughnesses $z_{0,obs}$. Cyan:$z_{0,1}$=0.001 m; magenta:$z_{0,1}$=0.01 m; orange:$z_{0,1}$=0.1 m; green:$z_{0,1}$=1 m.

the roughness uncertainty (in $\ln z_{0,1}$ space) may easily correspond to three or more times the reported (mean) $z_{0,1}$, one must integrate over $\ln z_{0,1}$ to find the relative uncertainty. Using (A1) and the substitution $x \equiv z_{obs}/z_{0,1}$ we have

$$\Delta \ln U_{pred}\Big|_{z_{0,1}}^{az_{0,1}} = \int_{z_{0,1}}^{az_{0,1}} \frac{\partial \ln U_2}{\partial \ln z_{0,1}} d\ln z'_{0,1} \simeq \int_{z_{0,1}}^{az_{0,1}} \frac{1.1}{z_{obs}} \frac{(z_{obs}/z'_{0,1})^{6/7}}{\ln(z_{obs}/z'_{0,1})} \frac{dz'_{0,1}}{(1 + z_{obs}/80\,\mathrm{m})^{1/7}}$$

$$= 1.1 \left(1 + \frac{z_{obs}}{80\mathrm{m}}\right)^{-1/7} \int_{z_{obs}/(az_{0,1})}^{z_{obs}/z_{0,1}} \frac{x^{-8/7}}{\ln x} dx$$

$$= \frac{1.1}{(1 + z_{obs}/80\mathrm{m})^{1/7}} \left\{ \mathrm{li}\left[\left(\frac{z_{obs}}{z_{0,1}}\right)^{-1/7}\right] - \mathrm{li}\left[\left(\frac{z_{obs}}{az_{0,1}}\right)^{-1/7}\right] \right\}. \tag{A2}$$



Here $a$ is the fractional uncertainty in observation-site background roughness as in (9), i.e. $a \equiv (z_{0,1} + \Delta z_{0,1})/z_{0,1}$ so that $\Delta z_{0,1} = (a-1)z_{0,1}$. The analytical logarithmic integral function $\mathrm{li}(x) \equiv \int_0^x (dt/\ln t)$ can be evaluated using typical contemporary mathematical programming libraries, scientific analysis programs, or via lookup-tables (Abramowitz and Stegun, 1972).[8]

### A2   Sensitivity to prediction-site roughness $z_{0,2}$

Just as above for the observation site background roughness, we can also express the uncertainty in predicted wind speed due to uncertainty in the roughness length for a prediction site. Following a similar procedure as above, using (7) and the substitution $y \equiv \ln z_{0,2}$ we obtain

$$
\begin{aligned}
\Delta(\ln U_{\mathrm{pred}})\Big|_{z_{0,2}}^{a z_{0,2}} &= \int_{\ln(z_{0,2})}^{\ln(a z_{0,2})} \frac{\partial \ln U_2}{\partial \ln z_{0,2}} d\ln z_{0,2}' \simeq \int_y^{y+\ln a} \frac{A + \ln(z_{\mathrm{pred}} f/G)}{[y' + \ln(f/G) + A](\ln z_{\mathrm{pred}} - y')} dy' \\
&= -\ln\left[\frac{A + \ln(f/G) + y'}{y' - \ln z_{\mathrm{pred}}}\right]\Big|_y^{y+\ln a} \\
&= \ln\left\{\left[1 - \frac{\ln a}{\ln(z_{\mathrm{pred}}/z_{0,2})}\right] \Big/ \left[1 + \frac{\ln a}{A - \ln[G/(f z_{0,2})]}\right]\right\}.
\end{aligned}
\tag{A3}
$$

### A3   Sensitivity to heights of measurement and prediction

Above it was written that predictions of wind speed (and thus AEP) were relatively insensitive to observation and measurement height, compared to the sensitivity to roughness. The minor dependence upon $z_{\mathrm{obs}}$ in (A2) and upon $z_{\mathrm{pred}}$ in (A3) is shown in Figure 10 for the case of grassland at measurement and observation sites ($z_{0,1}{=}z_{0,2}{=}4\,\mathrm{cm}$) as a function of roughness uncertainty in the form of $z_0$-bias. As one can see from the figure, the EWA method, i.e. via the geostrophic drag law, predicted $U$ has increased sensitivity to $\{z_{\mathrm{obs}}$ and $z_{\mathrm{pred}}\}$ for large uncertainties in roughness length (biases in Fig. 10). However, even for a bias $a_{\mathrm{bias}} \sim 3^{\pm 1}$ (+200% or −67%), the resultant $\langle U \rangle$-uncertainty spans a range smaller than −1% to 2%. For the case of independent uncertainties in $z_{0,1}$ and $z_{0,2}$, then the half-width of the associated $\Delta U$ distribution expands slightly, becoming roughly 3% for an 'input' roughness uncertainty ($\Delta z_0/\langle z_0 \rangle$ distribution half-width) of 3.

These height-induced uncertainty values are small enough that one could use Figure 8 for AEP-uncertainty (where $z_{\mathrm{obs}}{=}60\,\mathrm{m}$ and $z_{\mathrm{pred}}{=}100\,\mathrm{m}$), and approximate the effect of varying $\{z_{\mathrm{obs}}, z_{\mathrm{pred}}\}$ from $\{60\,\mathrm{m}, 100\,\mathrm{m}\}$ over simple terrain, by taking the difference between the curve for the desired $\{z_{\mathrm{obs}}, z_{\mathrm{pred}}\}$ and the $\{60\,\mathrm{m}, 100\,\mathrm{m}\}$ curve in Fig. 10, and multiplying this by the effective AEP($\langle U \rangle$) exponent $p$ (where the latter is detailed in the next appendix).

---

[8] The error-scaling function can also be written in terms of the exponential integral function $\mathrm{Ei}(x) \equiv -\int_{-x}^{\infty}(e^{-t} d\ln t)$, i.e. $\mathrm{Ei}(\ln x^{-1/7})$ evaluated at the same limits as in (A2).




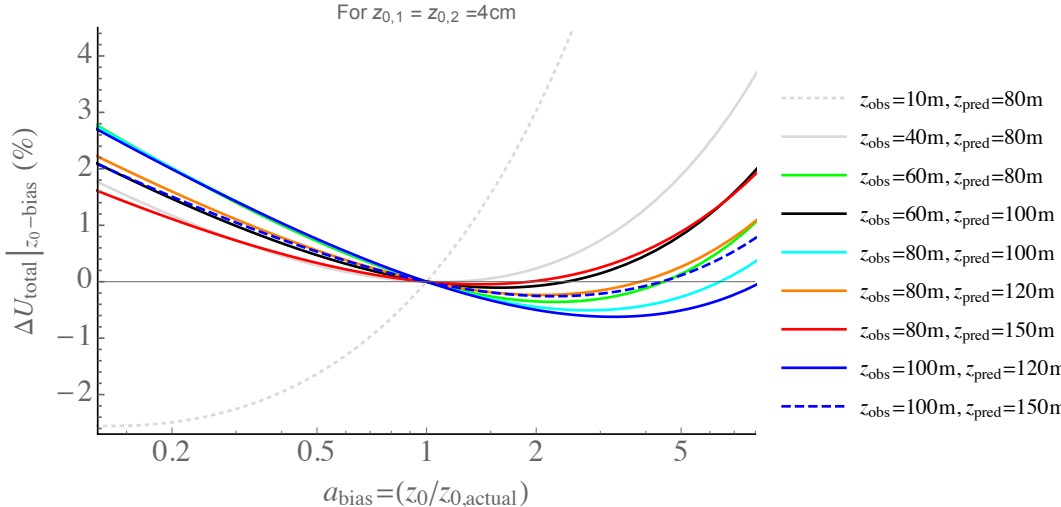

**Figure 10.** Total uncertainty versus bias in background roughnesses $z_{0,1}$ and $z_{0,2}$, due to different combinations of measurement and prediction heights, for the case of grassland ($z_0$=4 cm) at both measurement and prediction sites.

### Appendix B: Analytical power-curve forms for scalable calculation of AEP

To propagate the uncertainty in mean wind speed into the annual energy production (AEP), it is necessary to have a model for AEP in terms of mean wind speed. Assuming a Weibull distribution for wind speeds, we are able to relate the Weibull parameters to AEP, for a given power curve. In this appendix we produce a 'universal' power curve formulation, which allows us to derive an expression for conversion of Weibull-$A$ parameter (or mean wind speed) into AEP for any given turbine rated speed $V_{rat}$. The forms we provide here apply for wind speed distributions having Weibull-shape ($k$) parameter of roughly 2; such 'Rayleigh-distributed' mean winds tend to be the most commonly found (i.e. $k \approx 2$ tends to be most likely, c.f. Troen and Petersen, 1989; Kelly et al., 2014b).

A canonical form for power-curves including the smooth transition from 'ideal' to maximum power for mean winds approaching rated speed $V_{rat}$ is:

$$P(U/V_{rat}) = P_0 \times \left\{ \frac{1}{2} + \frac{1}{2}\tanh\left[\pi\left(U/V_{rat} - n^{-1/2}\right)\right] \right\}^n, \quad n = 3. \tag{B1}$$

We choose the 'order' $n$ to be 3, matching the ideal $P \propto U^3$ behavior in the regime for wind speeds above cut-in and below rated wind speed. Convolving (B1) with the Weibull probability density for wind speed

$$f(U) = \frac{k}{U}\left(\frac{U}{A}\right)^k \exp\left[-\left(\frac{U}{A}\right)^k\right] \tag{B2}$$



gives the normalized AEP, but this is not quite amenable to (simple) analytical relation. Thus in order to find a useful (closed) expression for the AEP, we make an approximation to the convolution $\int f(U)P(U)dU$ via (B1) and (B2):

$$\frac{AEP}{AEP_0} \simeq 0.3\left\{1 + \tanh\left[\pi\left(\frac{A}{V_{\text{rat}}} - \frac{1}{\sqrt{2}}\right)\right]\right\}. \tag{B3}$$

The closed-form approximation (B3) for normalized AEP is shown in Figure 11, along with the numerically integrated product of (B1) and (B2) which it approximates, for the case of Rayleigh-distributed wind speeds ($k$=2). The left-hand plot (Figure 11a) gives $AEP/AEP_0$ as function of mean wind speed $\langle U \rangle$[9] for a single value of $V_{\text{rat}}$, and also displays the results corresponding to use of either a simple $P \propto U^3$ power curve, or an ideally-limited power curve that has $P(U)/P_{\text{rat}} = \{(U/V_{\text{rat}})^3, 1\}$ for $\{U < V_{\text{rat}}, U \geq V_{\text{rat}}\}$. Figure 11b again shows the numerically integrated and approximated nominal power, but as a function of Weibull-$A$ and for different $V_{\text{rat}}$. One can see from Figure 11 that the approximation (B3) works well for mean wind speeds

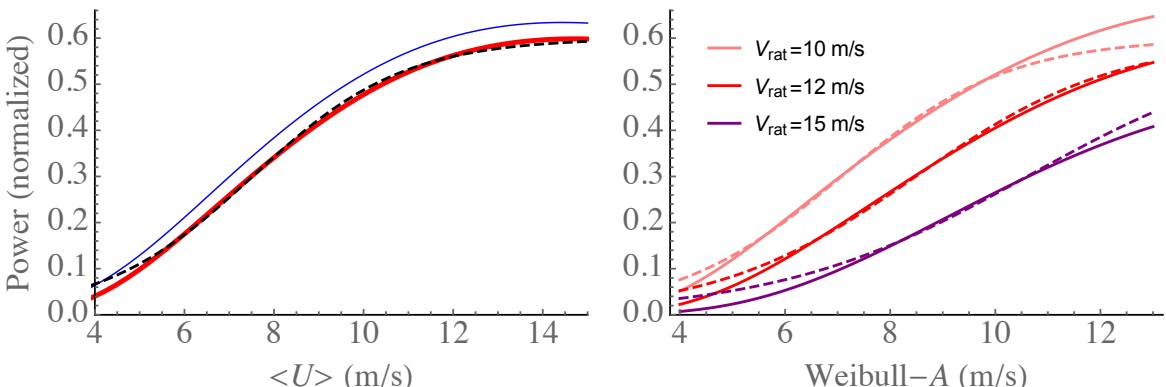

**Figure 11.** Left (a): normalized power versus mean wind speed, for $k=2$, $V_{\text{rat}}$=12 m s$^{-1}$; blue is for ideal truncated (sharp) power-curve, red is via numerically integrated 'universal' power-curve form (B1), black-dashed is approximation (B3) to universal form, and (green) dotted line is for simple $U^3$ form. Right (b): Normalized power (convolution of Eq. B1 and Weibull distribution) as a function of Weibull-$A$ parameter, for rated speeds of 10 m s$^{-1}$ (pink), 12 m s$^{-1}$ (red), and 15 m s$^{-1}$ (purple); dashed lines indicate analytic approximation as in (B3).

and rated speeds typical of multi-megawatt turbines (and associated hub-heights), i.e. $\langle U \rangle \sim$6–14 m s$^{-1}$ and $V_{\text{rat}} \sim$12–15 m s$^{-1}$. Most succinctly, given a Weibull-$A$ value (or mean wind speed) and turbine-rated speed $V_{\text{rat}}$, the AEP can be simply estimated by (B3) as a function of $A/V_{\text{rat}}$; this is shown in Figure 12a.

The effective wind-power exponent $p$ defined by $AEP = U^p$ can be now be found analytically from the corresponding analytical form (B3) for normalized AEP:

$$p = \frac{\ln AEP}{\ln\langle U \rangle} = \frac{\partial \ln AEP}{\partial \ln\langle U \rangle} = \frac{\pi(A/V_{\text{rat}})\text{sech}^2\left[\pi(A/V_{\text{rat}} - 2^{-1/2})\right]}{1 + \tanh\left[\pi(A/V_{\text{rat}} - 2^{-1/2})\right]}. \tag{B4}$$

The power-law exponent derived in (B4) is displayed in Figure 12b, for the case of Weibull-shape parameter $k = 2$. Evident from the figure is the optimal choice of sites having mean winds at hub-height that are $\sim$60–80% of rated speed, as well as the

---

[9] For Rayleigh-distributed wind speeds (Weibull, with $k = 2$), the mean wind is simply $\langle U \rangle = A\Gamma(1+1/k) \simeq 0.89A$.





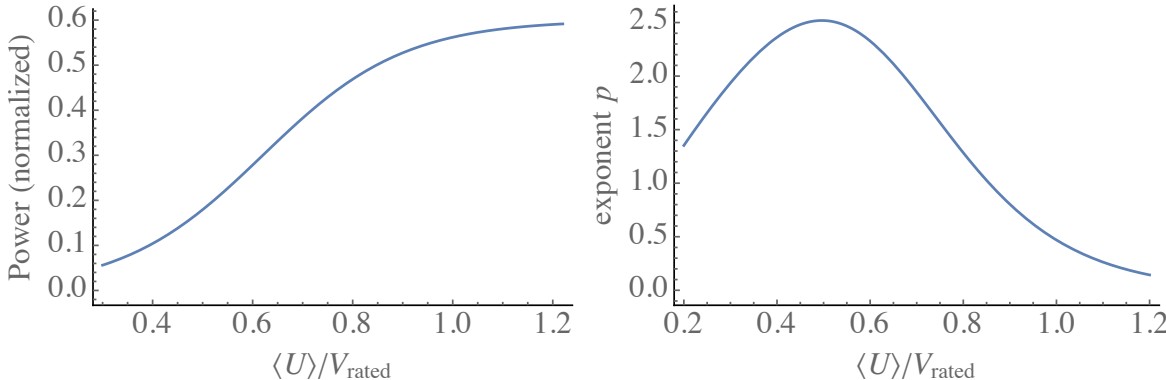

**Figure 12.** Left (a): Normalized AEP versus mean wind relative to rated wind speed. Right (b): AEP effective power-law exponent versus mean wind over rated speed, obtained via integrable power-law form (B1) and subsequent dimensionless AEP (B3) for Weibull-distributed wind with $k = 2$.

diminishing returns which can result from using turbines having rated speeds not much higher than the mean wind speed; this is consistent with common industrial practice, synthesized in e.g. the empirical study of Svenningsen (2015).

For a given value of $\langle U \rangle / V_{\mathrm{rat}}$, via $AEP \propto U^p$ and (B4), we are able to translate uncertainty in mean wind speed estimates (due e.g. to background roughness) into AEP uncertainty.

*Acknowledgements.* The authors would like to thank the reviewers for their time and effort towards constructive criticism of the present
5 article. MK is also grateful to Andrey Sogachev for discussion, and for pointing toward the Bosveld (1997) and Tian et al. (2011) works. MK further thanks Neil Davis for updates on logistic-function use in wind energy, and for trying out the analytical power-curve form on some big data sets.



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
