# Peer review of "Statistical characterization of roughness uncertainty and impact on wind resource estimation"

_Wind Energy Science, 2016_

## Referee Comment (RC1) · Anonymous Referee #1 · 29 Nov 2016

Authors have addressed an very interesting issue in this paper, i.e., the effect of roughness uncertainty on the prediction of wind speeds and annual wind energy production. The paper is technically sound. However, due to some very long complex sentences, it is very difficult to follow in some sections of the paper. A few of the long complex sentences are: - Page 15, Section 3.2.1: "Towards practical consideration . . . . . . . . . . . . . . . . . . . . . .off-shore predictions." - Page 17, Section 3.3.1: "The half-widths . . . . . . . . . . . . . . . . . . . .u* in the ASL." - Page 18, Section 3.3.2: " As mentioned. . . . . . . . . . . . . . . . . . . . . . . . . . . .by a factor p." Please break down these sentences and other long sentences into two or three short sentences. This will help readers to follow the paper easily. Minor corrections: - Page 2, line 12: add ',' after

Consequently - Page 2, line 16: remove 'annual energy production', use AEP only as it is already introduced - Page 2, line 26: 'some height', be specific - Page 3, line 16: 'and (i.e. du*2/dz«),', this is not clear - Page 7, line 20: use 'the' before 'observation-based zo' - Page 10, line 22: 'In terms of (9)', not clear. Equation (9)? - Page 17, line 32: remove 'e.g.' before direct LIDAR - Page 18, line 3-4: 'depends on such-most simply expressed via the exponent'-this part of the sentence is not clear - Conclusion is too big. Please summarize the key findings in the conclusion. Bullet points can be used.

---

## Author Comment (AC1) · 30 Nov 2016

The authors are grateful for the helpful comments of referee #1. We take the advice for reducing the length and complexity of a number of sentences which may be difficult to understand, and will submit a revision accordingly; this revision also includes the minor corrections suggested in this first comment/review.

The expression on line 16 of page 3 included a 'typo', and has been corrected to "...$u_*$ is effectively constant from the surface up to height $z$ (i.e. $du_*^2/dz \ll \sigma_u^2/\ell_{\mathrm{ABL}}$, Wyngaard 2010)."

Regarding the title of sub-section 3.1.1, "Uncertainty in observation-based $z_0$", I dis-

agree with the referee's suggestion to insert 'the' after 'in'. I respectfully decline, because the intent of the subsection is more general than simply covering just one dataset: we discuss uncertainty in $z_0$ as obtainable from measurements within the general framework given in the previous section, and through introduction of the metric $w_{\langle z_0 \rangle_{\mathrm{RS}}} / \langle \langle z_0 \rangle_{\mathrm{RS}} \rangle_g$ (dimensionless roughness-distribution width).

On page 10, line 22, we correct '(9)' to be equation '(4)'. We have also clarified the sentence on page 18 (before equation 11).

For the conclusion, we will attempt to make the revision's conclusion more concise, considering bullet-points as suggested.

---

## Referee Comment (RC2) · Anonymous Referee #2 · 4 Jan 2017

General Comments

The paper analyzes the uncertainty of turbine site wind speed predictions due to background surface roughness uncertainty. Wind turbine site predictions are based on the European Wind Atlas method (i.e. the geostrophic drag law) and surface roughness uncertainty is quantified from observations (assuming a logarithmic wind profile) and from an ensemble of wind engineer user inputs. Several approximations enable analytical expressions for the uncertainty of the predicted wind speed (and annual energy production) at the turbine site to uncertainties in the observation location surface roughness and turbine site surface roughness.

The authors have addressed an important issue of the wind energy community and the

approach is of high technical quality. In particular, careful use of mathematical approximations allowed to derive several analytical expressions which enable fairly simple and computationally inexpensive calculations of wind speed uncertainty. Overall the paper is well written although I agree with Referee #1 that some explanations are difficult to follow due to the use of long sentences.

Specific Comments

Would it be possible to assess the quality of the uncertainty prediction by using actual measurements at two sites?

Technical Corrections

P 3, L 17: Remove the parenthesis

P3, L 22: that is often used

P 5, L10: Reference is made to figure 1b but figure 1 has no labels called (a) or (b)

P 5, L 28: present article concerned with

P 6, equation (3): The meaning of $u_{(*0)}$ is not given (typo?)

P 6 , L 24: randomness inherent to the process(es)

P 8, L 1: ãĂŰ10ãĂŮˆ5 values of geometric-mean $z_0$ – Please you use the same symbol for this random variable that you use in equation (4)

P 17, L 17: flow over such surfaces?

P 21, L 32: inhomogeneity

---

## Author Response (AR1)

[revised manuscript text omitted]

**response to reviewer #1 (reviewer's comment 29 Nov 2016)**

The authors are grateful for the helpful comments of referee #1. We take the advice for reducing the length and complexity of a number of sentences which may be difficult to understand, and will submit a revision accordingly; this revision also includes the minor corrections suggested in this first comment/review.

5     The expression on line 16 of page 3 included a 'typo', and has been corrected to "...$u_*$ is effectively constant from the surface up to height $z$ (i.e. $du_*^2/dz \ll \sigma_u^2/\ell_{\mathrm{ABL}}$, Wyngaard 2010)."

    Regarding the title of sub-section 3.1.1, "Uncertainty in observation-based $z_0$', I disagree with the referee's suggestion to insert 'the' after 'in'. I respectfully decline, because the intent of the subsection is more general than simply covering just one dataset: we discuss uncertainty in $z_0$ as obtainable from measurements within the general framework given in the previous

10  section, and through introduction of the metric $w_{\langle z_0 \rangle_{\mathrm{RS}}}/\left\langle \langle z_0 \rangle_{\mathrm{RS}} \right\rangle_g$ (dimensionless roughness-distribution width).

    On page 10, line 22, we correct '(9)' to be equation '(4)'. We have also clarified the sentence on page 18 (before equation 11).

    As suggested for the conclusion, we will attempt to make it more concise, considering bullet-points.

**response to reviewer #2 (reviewer's comment 4 Jan 2017)**

We thank reviewer number 2 for their helpful comments.

15     Reviewer 2 suggested several technical corrections, which we have addressed:

- *P 3, L 17: "Remove the parenthesis"* We use/keep it, to avoid a run-on sentence.

- *P3, L 22: "that is often used"* ...'oft-used' is proper English usage (though a bit old).

- *P 5, L10: "Reference is made to figure 1b but figure 1 has no labels called (a) or (b)"*: Fixed in Figure 1 caption.

- *P 5, L 28: "present article concerned with"* Ok.

20   - *P 6, equation (3): "The meaning of $u_{*0}$ is not given (typo?)"* Removed '0'.

- *P 6 , L 24: "randomness inherent to the process(es)"* We respectfully disagree; 'inherent in' is also acceptable English usage (Oxford English Dict.).

- *P 8, L 1: "...$10^5$ values of geometric-mean $z_0$ – Please you use the same symbol for this random variable that you use in equation (4)"* Fixed.

25   - *P 17, L 17: "flow over such surfaces?"* Changed to 'flow over such terrain'.

- *P 21, L 32: "homogeneity"?* Fixed: prepended 'in' to 'homogeneity'.

    Reviewer 2 also asked, *"Would it be possible to assess the quality of the uncertainty prediction by using actual measurements at two sites?"*

    Yes it would, but this is beyond the scope of the current article. Such validation requires a pair of sites within the same wind

30  climate, i.e. having the same geostrophic wind; the data of the study came from a site that did not have such a 'partner'-site.